# Improving recombinant protein production by yeast through genome-scale modeling using proteome constraints

Feiran Li [1], Yu Chen [1,6], Qi Qi [1,6], Yanyan Wang[1,6], Le Yuan [1,2], Mingtao Huang[1,4], Ibrahim E. Elsemman[1,5], Amir Feizi [1 ✉], Eduard J. Kerkhoven [1,2] & Jens Nielsen [1,3 ✉]

Eukaryotic cells are used as cell factories to produce and secrete multitudes of recombinant pharmaceutical proteins, including several of the current top-selling drugs. Due to the essential role and complexity of the secretory pathway, improvement for recombinant protein production through metabolic engineering has traditionally been relatively ad-hoc; and a more systematic approach is required to generate novel design principles. Here, we present the proteome-constrained genome-scale protein secretory model of yeast *Saccharomyces cerevisiae* (pcSecYeast), which enables us to simulate and explain phenotypes caused by limited secretory capacity. We further apply the pcSecYeast model to predict overexpression targets for the production of several recombinant proteins. We experimentally validate many of the predicted targets for α-amylase production to demonstrate pcSecYeast application as a computational tool in guiding yeast engineering and improving recombinant protein production.

[1] Department of Biology and Biological Engineering, Chalmers University of Technology, Kemivägen 10, SE412 96 Gothenburg, Sweden. [2] Novo Nordisk Foundation Center for Biosustainability, Chalmers University of Technology, Kemivägen 10, SE-412 96 Gothenburg, Sweden. [3] BioInnovation Institute, Ole Måløes Vej 3, DK2200 Copenhagen N, Denmark. [4] Present address: School of Food Science and Engineering, South China University of Technology, Guangzhou 510641, China. [5] Present address: Department of Information Systems, Faculty of Computers and Information, Assiut University, Assiut, Egypt. [6] These authors contributed equally: Yu Chen, Qi Qi, Yanyan Wang. ✉email: afeizi@gmail.com; nielsenj@chalmers.se

The protein secretory pathway is an important pathway for eukaryotic cells. Numerous native proteins are processed by the secretory pathway in eukaryotes; around 10–20% in fungal species[1,2] and 30–40% in mammals[3]. The secretory pathway spans several different organelles carrying out peptide translocation, folding, Endoplasmic reticulum (ER)-associated protein degradation (ERAD), sorting processes as well as different post-translational modifications (PTMs), ensuring proper protein functionality[4]. There are around 200 proteins engaged in the protein secretory pathway in yeast *Saccharomyces cerevisiae*, hence responsible for these functions. The specific PTM profile of each secretory protein dictates which specific combination of multiple processes is required for its production and secretion. This makes the secretory pathway a complicated production line and therefore complex to describe. It is therefore desirable to unravel the energetic costs for processing proteins passing through the secretory pathway, and how the cell distributes energy and enzymes to process these proteins, as this would facilitate a better understanding of protein secretion.

*S. cerevisiae* is used as expression system for roughly 15% of all protein-based biopharmaceuticals for human use on the market[5]. It has also been used as an important model organism for studying this important pathway, and many discoveries made in yeast translate directly to other eukaryotes, such as Chinese Hamster Ovary (CHO) cells that are also widely used for the production of protein-based biopharmaceuticals[6,7]. Since the early days of recombinant protein production in the 1980s, there have been many attempts to improve the protein expression and secretion levels by removing bottlenecks in the protein modification and secretion pathway[8]. However, most of these attempts were evaluated for one recombinant protein only, and often identified targets do not translate into the improved expression of another protein. Furthermore, the protein yield has typically been much lower than the theoretically estimated range[9]. There is therefore much interest in developing a rational design tool for optimization of the secretory pathway for any recombinant protein, in line with what has been developed for metabolism in many cell factories[10].

There are several published frameworks or models for describing protein secretion in yeast and other eukaryotes, but they are either not able to perform simulations or contain only a partial description of the protein secretory pathway[4,11–14]. Even for a recently published secretory model for mammalian cells[13], the model is solely a basic extension of a genome-scale metabolic model (GEM), which is not able to simulate how native secretory proteins compete with recombinant proteins targeted to pass through this pathway. Besides that, even though engineering targets have been predicted using basic GEMs for recombinant protein overproduction[14–16], those targets are related to metabolism without the investigation of the protein secretory pathway due to the nature of basic GEMs.

In this work, we reconstruct a detailed proteome-constrained genome-scale protein secretory model for *S. cerevisiae* (pcSecYeast). This model contains a description of the complete protein secretory pathway and can perform multiple types of simulations including the competition between recombinant and native secretory proteins. The model also enables calculation of the energetic cost for native secretory proteins and hereby enables investigation on how misfolded proteins cause growth reduction. We use the model to evaluate the secretion of various recombinant proteins and predict engineering targets for improving their production. The model represents a significant advancement in terms of enabling more rational design of yeast cells to be used for recombinant protein production, while furthermore providing a scaffold for building similar models for other eukaryotic cells, e.g., CHO cells.

## Results

**Construction of pcSecYeast**. We first updated the latest yeast GEM Yeast8[17] by adding 92 metabolic reactions to enable the synthesis of precursors required in the secretory pathway such as glycosylphosphatidylinositol (GPI) anchor and glycans (Supplementary Data 1). Similar to the metabolic-expression (ME) model for *Escherichia coli*[18] and *S. cerevisiae*[19], protein expression, translation, folding, and degradation were subsequentially added for all proteins in the model. Additionally, for proteins processed in the secretory pathway, we added reactions that comprehensively describe protein processing, including translocation, post-translational modification, folding, misfolding, complex formation and degradation (Fig. 1a). Hereby the model describes all detailed processes from nascent peptide in the cytosol to the final mature form in their destination compartment for each protein in the model. Therefore, pcSecYeast adds a much more comprehensive description of protein translocation and processing compared with earlier ME models. A comparison of pcSecYeast with relevant models for *S. cerevisiae*[19–21] and other secretory models[13,14] is available in Table 1 (detailed information in Supplementary Method 1). To our knowledge, pcSecYeast represents the model to describe close links between metabolism, protein translation, post-translational protein processing, protein degradation, and protein secretion in yeast and can be easily adapted to other cell types. The components that participate in the protein secretory pathway are involved in 12 subsystems (Fig. 1b). Overall, pcSecYeast accounts for 1639 protein-coding genes (1156 metabolic genes and 483 protein synthesis- and secretion-related genes) and approximately 70% of the total proteome mass (45.7% from metabolic proteins, 20.6% related to ribosome, proteosome and secretory machinery proteins and 4.6% from unmodeled secretory proteins) according to PaxDb[22] (Supplementary Data 2). Details of the reconstruction process and parameter collection can be found in the Supplementary Method 2–6. All reactions and metabolites of pcSecYeast can be found in the Supplementary Data 3-4.

As an extension of Yeast8, pcSecYeast includes default constraints such as mass conservation and flux bounds on metabolic reactions. In addition, we introduced coupling constraints to relate protein synthesis with metabolism (Supplementary Method 6). The metabolic part in the model supplies the substrate and energy for the protein-related part, such as ribosome and enzyme synthesis, while the metabolite conversion processes in the metabolic part are catalyzed by enzyme complexes synthesized in the protein-related part (Fig. 1c). Protein synthesis is constrained by the synthesis of ribosome and other machineries, such as secretory machinery complexes (Fig. 1c). Each metabolic flux in the model is constrained by the maximal capacity of the associated enzyme, which is a function of turnover rate ($k_{cat}$) and the enzyme concentration. Thus, we can simulate the minimum protein levels which sustain the metabolic state, i.e., the proteome-constrained metabolic state. This means that the proteome composition in pcSecYeast is not a fixed amount of average amino acid compositions as in the basic GEMs, but a dynamically changing composition of enzymes, which reflects the cell state at a certain condition. Thus, the model enables simulating cellular resource allocation under different conditions, such as how the cell would balance recombinant protein with native secretory proteins in the recombinant protein production and how the cell would optimize its enzyme profile among various environmental conditions.

**Secretory cost correlates with the switch of hexose transporters**. Transporters are one important group of proteins that pass through the secretory pathway. Yeast has multiple hexose transporters with diverse kinetics, which are expressed at different

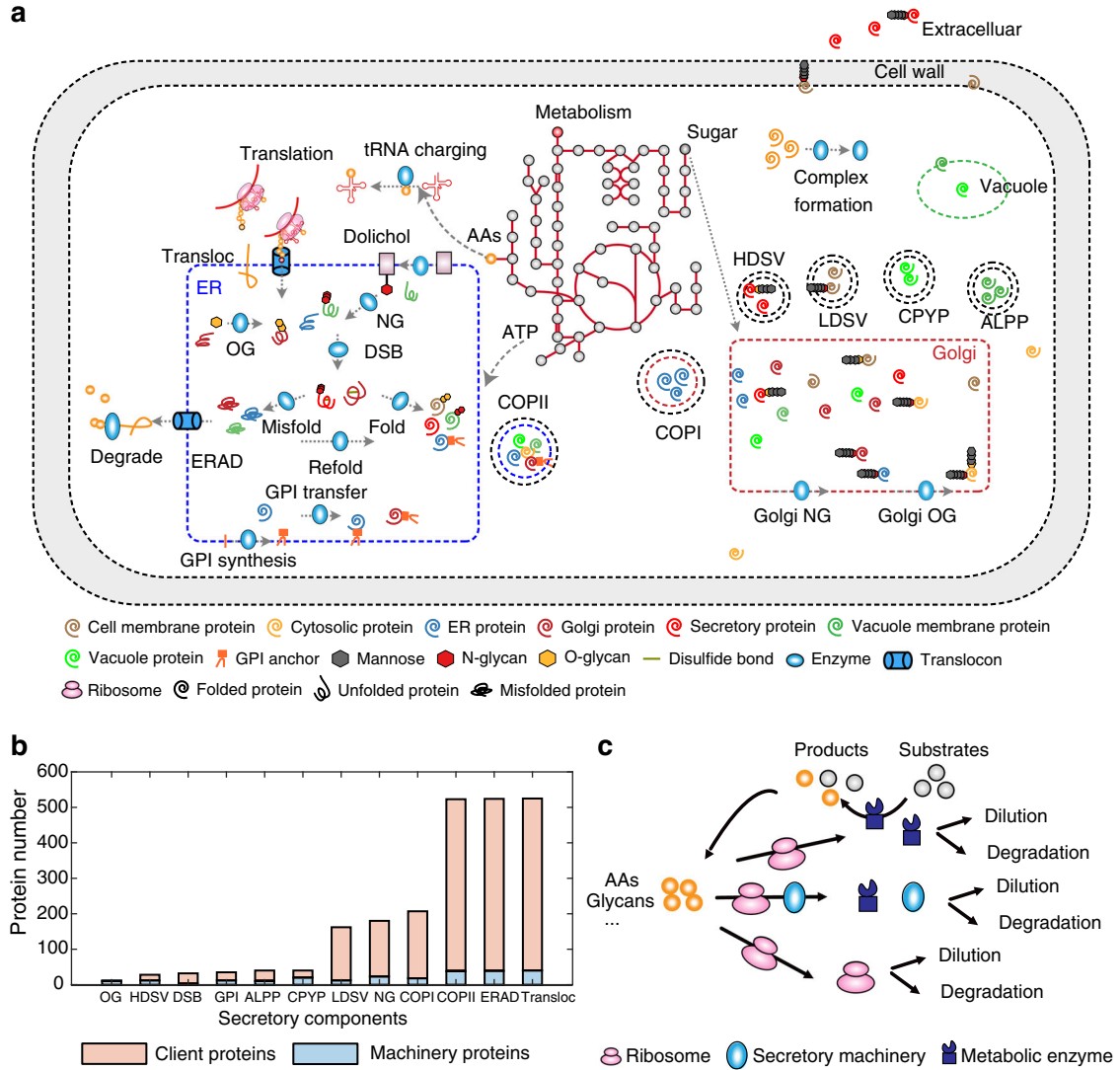

**Fig. 1 Overview of components in pcSecYeast. a** Simplified schematic processes involved in the protein secretory pathway. The process includes protein translation, translocation, glycosylate, GPI transfer, ERAD and sorting process. The detailed description of all components and reactions can be found in Supplementary Methods. The protein secretion reactions are added for all proteins in the model that are processed in the secretory pathway based on their PTM profiles and amino acid sequences. Transloc translocation, NG *N*-glycosylation, OG *O*-glycosylation, DSB disulfide bond formation, GPI glycosylphosphatidylinositol, ER Endoplasmic reticulum, ERAD ER-associated degradation, LDSV low-density secretory vesicles, HDSV high-density secretory vesicles, ALPP alkaline phosphatase pathway, CPYP carboxypeptidase Y pathway. **b** Subsystems in the secretory pathway and the protein number in each subsystem. All proteins processed by the secretory pathway are client proteins for Transloc, COPII and ERAD. **c** Coupling process in the model. The metabolic part produces energy and precursors such as amino acids, glycans for enzyme and ribosome synthesis. Enzymes constrain these metabolic reactions. Ribosomes constrain protein translation. The secretory machinery constrains protein processing in this pathway. All proteins, including ribosomes are diluted due to growth and degraded due to misfolding. Source data are provided as a Source Data file.

levels under different extracellular glucose concentrations[23]. The benefit of utilizing high-affinity transporters during nutrient depletion or limited conditions seems evident, but questions remain on why the cell would switch to low-affinity transporters[24]. To investigate the switch, we utilized pcSecYeast to simulate yeast growth under different glucose concentrations. As a result, the model captured the metabolic shift referred to as the Crabtree effect, i.e., the production of ethanol at high specific growth rates (Fig. 2a). Furthermore, the model correctly predicted a switch from the predominant use of the high-affinity glucose transporter (Hxt7) to low-affinity glucose transporters (Hxt3 and Hxt1) at high glucose concentrations (Fig. 2b), which is consistent with the experimental observation that *HXT3* and *HXT1* genes are only expressed at high specific growth rates[23]. Using the model, we can calculate the secretory cost of utilizing sole specific glucose

transporter at corresponding conditions. The calculation is illustrated by Eq. (1). The secretory cost can be calculated as the required abundance of the transporter multiplied by the unit secretory cost. The protein abundance of the transporter $[E_i]$ is determined by the total glucose uptake rate $V_{glc}$, $K_M$ and extracellular glucose concentration $[S]$ according to the Michaelis-Menten equation. The unit secretory cost is defined as the cost required for translation, modification, and secretion of one mol specific protein, which can be predicted by pcSecYeast (Methods). We predicted the unit secretory costs for all native secretory proteins in *S. cerevisiae* (Supplementary Data 5) and found that Hxt1 has a relatively lower unit secretory cost compared to Hxt7, suggesting that synthesizing one mol Hxt1 would pose less energy burden on the cell. This is partly because Hxt1 has fewer

**Table 1 Comparison of pcSecYeast with other models.**

| Models | ihGlycopastoris | Mammalian secretory model | pcYeast | yETFL | pcSecYeast | WM_S288C |
|---|---|---|---|---|---|---|
| Reference | 14 | 13 | 21 | 19 | This study | 20 |
| Model type | Basic GEM | Basic GEM | Fine-grained pcGEM | Fine-grained pcGEM | Fine-grained pcGEM | whole-cell model |
| Organism | P. pastoris | Mammalian cells | S. cerevisiae | S. cerevisiae | S. cerevisiae | S. cerevisiae |
| Model assumption | Steady state | Steady state | Steady state | Steady state | Steady state | Dynamic |
| Constraint | Mass balance | Mass balance | Mass balance, kinetic, volume constraint | Mass balance, kinetic, thermodynamic constraints | Mass balance, kinetic constraint | - |
| Metabolism# | Yes | Yes | Yes | Yes | Yes | Yes |
| Transcription# | No | No | No | Yes, lumped | No | Yes |
| Translation# | * | * | Yes | Yes | Yes | Yes |
| Folding# | * | * | Lumped | Lumped | Comprehensive | Lumped |
| Misfolding# | No | * | Lumped | Lumped | Comprehensive | Lumped |
| Post-translational modification# | NG | NG, OG, DSB | No | No | NG, OG, GPI and DSB | Phosphorylation, acetylation and ubiquitination |
| Degradation# | * | * | Lumped | Lumped | Comprehensive | Lumped |
| Ribosome assembly# | No | No | Yes | Yes | Yes | Yes |
| Simulate proteome changes | No | No | Yes | Yes | Yes | N/A |
| Simulate protein misfolding | No | No | No | No | Yes | N/A |
| Simulate native protein competition with recombinant protein | No | No | N/A | N/A | Yes | N/A |
| Predict engineering targets for improving recombinant proteins | Only targets in metabolic pathway | Only targets in metabolic pathway | N/A | N/A | Targets both in secretory and metabolic pathways | N/A |

# stands for process coverage, * means the process is added only for the recombinant protein, not for native proteins, N/A means that the description is not applicable for the specific model. GEM Genome scale metabolic model, pcGEM proteome constrained genome-scale metabolic model, P. pastoris Pichia pastoris, S. cerevisiae Saccharomyces cerevisiae, NG N-glycosylation, OG O-glycosylation, DSB disulfide bond formation, GPI glycosylphosphatidylinositol.

N-glycosylation modification sites than Hxt7 (Supplementary Data 6). Combining the unit secretory cost with the total glucose uptake rate, extracellular glucose concertation, $k_{cat}$, and $K_M$, we can calculate the secretory cost for utilizing each glucose transporter at different specific growth rates using Eq. (1) (Fig. 2c). The calculated secretory cost suggests that utilization of Hxt1 and Hxt3 would gradually gain the advantage over Hxt7 with increasing glucose concentrations (Fig. 2c). The switch of cost perfectly aligns with the experimentally observed switch of glucose transporters, which serves as an explanation for the transporter switch. We also performed sensitivity analysis on the $k_{cat}$ for Hxt1 and found that even if we set the $k_{cat}$ for Hxt1 at the same value as Hxt7, Hxt1 would still be favorable for glucose uptake in the model simulation at the maximum specific growth rate (Supplementary Fig. 1). This suggests that the slightly lower unit secretory cost of Hxt1 may contribute to the transporter switch, particularly at the proteome-constrained conditions at high specific growth rates. Our model hereby predicts that the switch of different affinity glucose transporters may be explained by the resource optimization strategy of the cell to adapt to limited resources.

**Yeast suppresses expression of high-cost secretory proteins under secretion pressure.** The protein secretory pathway is concurrently processing hundreds of proteins that compete for limited resources such as energy, precursors, and components of the secretory machinery. It has been reported that recombinant mammalian cells repress the expression of native energetically expensive secretory proteins to save limited resources for growth and recombinant protein production[13]. With our proteome allocation model of the secretory pathway, we can perform not only the same calculation of the costs of all 497 native secretory and cell membrane proteins in yeast as done for mammalian cells[13] (denoted as direct cost in the Supplementary Fig. 2a) but also a more accurate analysis of the costs including the additional costs for corresponding shares of catalyzing enzymes and secretory machineries required for processing the protein besides the cost for itself (unit secretory cost in Supplementary Fig. 2a). By correlating unit secretory cost with direct cost, we found that the unit secretory cost calculated in pcSecYeast is overall 3.5-fold higher than the direct cost (Supplementary Fig. 2a). Outliers in the correlation of these two types of cost calculation are mainly caused by unusual protein features such as the 52 N-glycosylation sites annotated for the protein Rax2 or long amino acid sequences for large proteins Tor1 and Tor2 (Supplementary Fig. 2a). To evaluate whether there is reduced expression of proteins that are costly to process by the secretory pathway, as observed in mammalian cells, we correlated the calculated unit secretory costs with the mRNA levels of 497 native secretory proteins for three strains with different levels of recombinant α-amylase production that were characterized in a recent study[25]. We observed a significant negative correlation (Pearson correlation coefficient < −0.27, P value < 1e-8) between unit secretory cost and mRNA level of native secretory proteins for α-amylase production strains (Supplementary Fig. 2b for MH34 and Supplementary Fig. 2c for all three strains), suggesting that the cells suppress the expression of proteins that are expensive to secrete when the secretory pathway is under pressure to process a recombinant protein. Moreover, we found that the negative correlations are stronger in the strains with higher α-amylase production levels (MH34 and B184) compared with that in a strain with a lower α-amylase production level (AAC) (Supplementary Fig. 2c, P value = 0.004). Therefore, the suppression level for costly native secretory proteins depends on the recombinant protein production levels, suggesting that the yeast cells respond accordingly to the level of secretion stress.

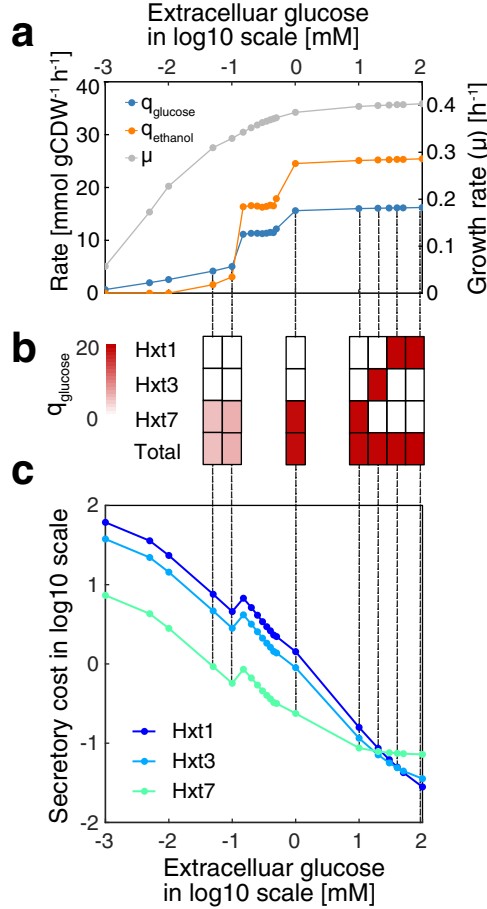

**Fig. 2 Simulated physiological response of _S. cerevisiae_ as a function of the extracellular glucose concentration. a** Simulated glucose uptake rates, ethanol production rates and specific growth rates under different extracellular glucose concentrations. Each point is the simulated result under a certain extracellular glucose condition. **b** Specific glucose uptake rate carried by each glucose transporter and the total glucose uptake rate. Hxt1 and Hxt3 are two low-affinity glucose transporters, while Hxt7 is a high-affinity glucose transporter. **c** Calculation of secretory costs of different glucose transporters with the total glucose uptake rate at input for each extracellular glucose concentration, unit secretory cost, $K_M$ and $k_{cat}$ that are specific to each transporter based on Eq. (1) in the Methods. Unit secretory cost is independent from the extracellular glucose concentrations and glucose uptake, while the secretory cost (y-axis) represents the cost for utilizing specific glucose transporters to sustain specific glucose uptake and the corresponding growth rate, respectively, which is a combination result of enzyme kinetics and total glucose uptake rate as described in Eq. (1). Source data are provided as a Source Data file.

**Misfolded protein slows maximum growth.** Protein synthesis and secretion is an error-prone process. Mutation in the sequence, errors during the synthesis or environmental stress cause the newly synthesized protein to misfold[26]. Misfolded proteins are prioritized to be rapidly eliminated by the ERAD pathway, but may be retained and accumulated in the ER, potentially triggering cell stress (Fig. 3a)[27–30]. Here, we used our model to simulate the ER tolerance to misfolded proteins. We expanded pcSecYeast to include the production of vacuolar carboxypeptidase Y (YMR297W, CPY), since CPY and its derived misfolded form CPY* are processed in the secretory pathway, and widely used in the elucidation of the mechanisms of ER quality control and ERAD of misfolded proteins[31]. By modifying the misfolding-ratio parameter in the model, we can simulate various

levels of CPY misfolding. A misfolding ratio of 100% means that all the CPY protein molecules are misfolded and cannot be targeted to the Golgi for further processing, representing the misfolded form CPY* as reported in literature[32].

Here, we used the maximum growth rate reduction to indicate the fitness cost of CPY going through different routes: 1) all correctly folded and targeted to the vacuole without misfolding; 2) misfolded in different ratios and some targeted for ERAD (here we use 45% misfolding ratio to represent the native degradation ratio[33] and 100% misfolding ratio for fully misfolded form CPY*); 3) all misfolded and retained in the ER for different times. Our simulations showed that misfolding imposes more fitness cost compared with correct folding; that retention imposes more fitness cost compared with ERAD; and that retention in the ER for a longer time would also impose more fitness cost (Fig. 3b). The model predicted that a lower level of misfolded CPY (native level CPY expression, 100% misfolded) has a smaller impact on cell growth. However, when misfolded CPY is expressed in larger amounts (25-fold CPY expression, 100% misfolded), there is a higher fitness cost. The simulation is consistent with experimental observations[32].

If the misfolded proteins are degraded by ERAD and the proteasome, then amino acids and modification precursors such as glycans can be recycled. However, if misfolded proteins are retained in the ER, they would compete with unfolded proteins for limited ER quality control machineries especially Kar2 and Pdi1[32], which would lower the processing rate of correctly folded proteins and increase the ER burden. We investigated the simulated various protein levels and found that the levels of Kar2 and Pdi1 increase significantly when CPY is retained (Supplementary Fig. 3), which suggests that the retained protein would drain Kar2 and Pdi1 and therefore compete with native proteins processed in the secretory pathway. In addition, we evaluated the ER redox stress by comparing the transport of glutathione (GSH) and glutathione disulfide (GSSG) and found that the flux of GSSG export from the ER is significantly higher when misfolded protein is retained in the ER (Supplementary Fig. 4), suggesting the higher redox unbalance in the ER at this state. The simulated transport increase is also in line with experimental observations[34].

Furthermore, we performed analyses to identify parameters leading to misfolded protein accumulation in the ER (Supplementary Fig. 5a–d, Fig. 3c). When retro-translocation enzymes (Doa10 and Hrd1 complexes) were constrained, the excessive misfolded CPY would be retained and accumulated in the ER when CPY was expressed at high levels, causing a steeper decrease in the specific growth rate (Fig. 3c). Other parameters such as ERAD capacity, ER volume, ER membrane space, and secretory machinery capacity were not able to show the retention and accumulation phenotype when constrained in the model (Supplementary Fig. 5a–d). We found that the retention of the misfolded protein phenotype is alleviated when removing the constraint of retro-translocation enzymes, suggesting the importance of retro-translocation toward handling of misfolded proteins (Supplementary Fig. 5e). Therefore, we can use pcSecYeast with the extra constraint on retro-translocation enzymes to mimic various states of misfolded protein accumulation in the ER (Fig. 3c). The plateau in the CPY degradation rate demonstrates that there is a maximum capacity of the retro-translocation and therefore also a tolerance limit for misfolded CPY.

**Protein features impact recombinant protein production.** Different secretory proteins are processed by different components of the secretory pathway based on their amino acid

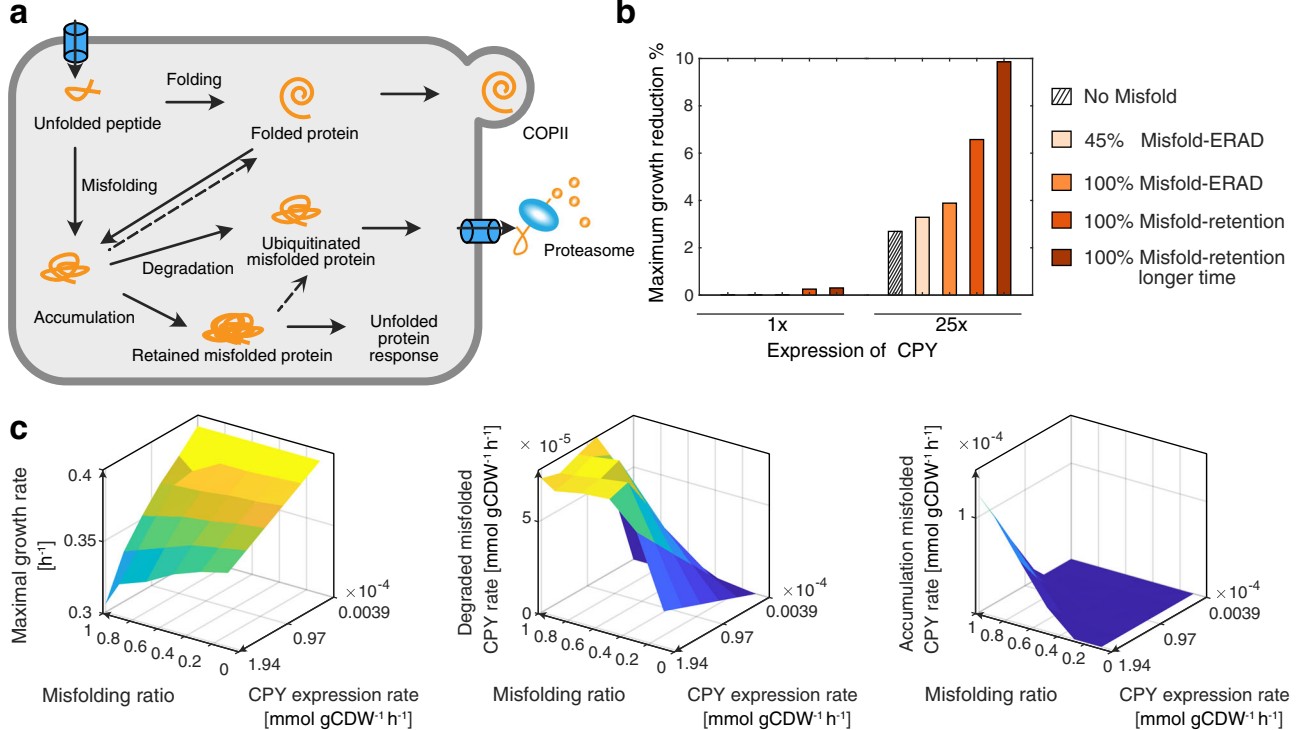

**Fig. 3 Simulation of CPY overexpression. a** Schematic view of different routes for expressed CPY. Processes represented in dotted lines are not considered in the pcSecYeast. **b** Reduction of simulated maximum specific growth rate due to expression at specific levels of CPY following different routes. 1x means its native expression level, 25x means 25-fold of its native expression level. Native expression level of CPY is from PaxDb database[22]. **c** Simulations for various CPY expression levels and misfolding ratios with the constraint for retro-translocation enzymes. Source data are provided as a Source Data file.

composition and PTMs. To identify the factors that influence secreted protein levels, we expanded pcSecYeast to describe the production of eight different recombinant proteins by adding the corresponding recombinant protein production and secretion reactions, respectively. These eight recombinant proteins differ in protein size and PTMs (Fig. 4a, detailed information in Supplementary Data 7). Note that hemoglobin folds with heme as a prosthetic group, which requires balancing of heme biosynthesis and its recombinant protein production (Fig. 4a)[35]. We generated eight specific models to simulate the maximum recombinant protein secretion under various growth rates. We observed that the maximum production rates were achieved at low specific growth rates for all the studied recombinant proteins (Fig. 4b), consistent with previous reports of bell shape kinetics for recombinant protein production in *S. cerevisiae* and *Pichia pastoris*[36–40]. Insulin precursor (IP) and α-amylase production were reported as growth-dependent[41], but only for the investigation of a more narrow interval of specific growth rates (0.05-0.2 h$^{-1}$), which is consistent with the model simulations. At high specific growth rates, there is a clear drop of production rate for all recombinant proteins (Fig. 4b), which clearly shows that at high specific growth rates the cell prioritizes its limited capacity of the secretory pathway to native proteins. It is important to note that a basic GEM can only describe a linear negative correlation of recombinant protein production with increasing specific growth rates (Supplementary Fig. 6). Moreover, the fact that the simulated α-amylase production by the basic GEM is around 30 times higher than experimental values[42], even with the measured glucose uptake rate as a constraint, highlights that basic GEMs are unfit for recombinant protein simulation (Supplementary Fig. 6).

We additionally investigated which protein feature influences recombinant protein production the most through a parameter importance analysis by machine learning. We found that PTMs on average have a higher impact on recombinant protein production compared with amino acid composition (Fig. 4c, Supplementary Fig. 7 for fivefold cross validation). Among all simulated features, *O*-glycosylation and *N*-glycosylation have larger negative impacts on recombinant protein production, which suggests that having more glycosylation sites would cause more burden for the cell (Fig. 4c).

**FSEOF identifies overexpression targets for recombinant protein overproduction**. Identifying engineering targets is crucial to improve the specific recombinant protein production rate. Predicting gene overexpression targets is more difficult and complex than predicting gene deletion targets since amplification of gene expression does not always increase the metabolic fluxes[43]. To fully validate the predictive power of pcSecYeast, we used the generated recombinant protein-specific models to predict overexpression targets for increasing recombinant protein production. Target prediction was performed using adapted Flux Scanning based on Enforced Objective Function (FSEOF)[43], where the model was constrained with a stepwise decrease in the specific growth rate, and recombinant protein production was maximized. The original FSEOF method selects fluxes that increase with the enforcement of recombinant protein production in the GEM simulations and identifies those reactions and associated genes as overexpression targets. Since we can compute the protein levels from the pcSecYeast simulations, we can directly select proteins, as overexpression targets, whose increased levels would result in increased recombinant protein production (Fig. 5a and Supplementary Data 8–15 for prediction results of these eight recombinant proteins). The predicted overexpression targets were

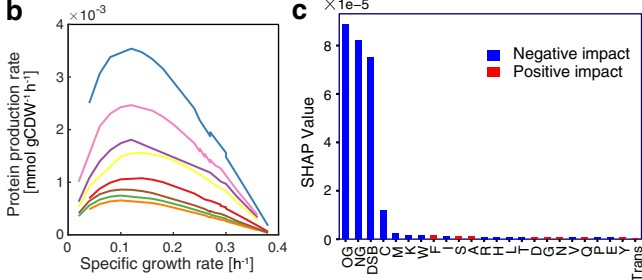

**a**

| Protein | abbr. | DSB | NG | OG | GPI | Length |
|---|---|---|---|---|---|---|
| Insulin precursor | IP | 3 | 0 | 0 | 0 | 53 |
| Human granulocyte colony stimulating factor | hGCSF | 2 | 0 | 1 | 0 | 174 |
| Hemoglobin | Hemoglobin | 0 | 0 | 0 | 0 | 299 |
| β-glucosidase | BGL | 0 | 0 | 0 | 0 | 421 |
| α-amylase | α-amylase | 4 | 1 | 0 | 0 | 478 |
| Acid phosphatase | PHO | 8 | 9 | 0 | 0 | 435 |
| Human serum albumin | HSA | 17 | 1 | 0 | 0 | 585 |
| Human transferin | HTF | 19 | 0 | 1 | 0 | 679 |

**Fig. 4 Simulation of recombinant protein production. a** Overview of protein features for eight recombinant proteins produced by *S. cerevisiae*. See Supplementary Data 7 for detailed information. Abbr. abbreviation. **b** Simulation of maximum specific recombinant protein production rate as a function of specific growth rate. **c** Feature importance analysis towards recombinant protein production. NG *N*-glycosylation site, OG *O*-glycosylation site, DSB disulfide bond number, Trans transmembrane domain, single letters stand for specific amino acids, SHAP value SHapley Additive exPlanations value. Fivefold cross validation was performed to validate the result (Supplementary Fig. 7). Source data are provided as a Source Data file.

ranked with priority scores and compared among the eight recombinant proteins (Fig. 5b, c). We predicted average 117 overexpression targets for each of the eight recombinant proteins with the majority of them (80%) being in the secretory pathway and 20% in the metabolic part of the model (Fig. 5b, c). The identified targets were more likely shared by recombinant proteins when they have the same PTMs. For example, targets in the *O*-glycosylation pathway were shared by *O*-glycosylated human-transferrin (HTF) and human granulocyte colony-stimulating factor (hGCSF) (Fig. 5c). Surprisingly, even though insulin precursor (IP) contains no *N*-glycosylation site, some predicted overexpression targets are related to *N*-glycosylation. This is explained by the fact that *N*-glycosylation is required for some secretory machinery proteins such as Pdi1 which catalyzes disulfide bond formation in IP production. By removing the disulfide bonds in IP, we found that those *N*-glycosylation-related genes were no longer predicted as targets (Supplementary Data 16). There are 41 predicted targets shared by all eight proteins, which are mainly involved in sorting, ER-Golgi transport and translocation from cytosol to the ER, suggesting the general importance of these processes in protein secretion (Fig. 5c). We also showed that hemoglobin is the recombinant protein with multiple unique targets in metabolism, especially for heme production, which demonstrates that metabolism is equally important along with the secretory pathway for improving hemoglobin production. For all other recombinant proteins, the secretory pathway is more limiting according to the prediction.

**Experimental validation for predicted α-amylase targets.** We next validated the predicted overexpression targets for improved α-amylase production. The 116 predicted overexpression targets for α-amylase overproduction were grouped by their function, of

which 28 were from metabolism and 88 were from the secretory pathway (Supplementary Fig. 8a). We selected 18 targets with different functions for further validation, most of them are with high priority scores (Supplementary Fig. 8a, b). There were 14 targets in the secretory pathway spanning translocation, folding, protein quality control, and sorting subsystems, and four targets in the metabolic part of the model, which are related to *N*-glycan synthesis and amino acid synthesis (Fig. 6a).

We next sought to test if individual overexpression of the predicted secretory targets could improve the α-amylase production rate. Among them, the glucosidase Cwh41[25], COPII-coated vesicles proteins Erv29[44], Sec16[45] and protein disulfide isomerase Pdi1[44,46] have already been validated, i.e., overexpression of these proteins can improve α-amylase production and secretion.

As for the remaining ten secretory targets, we performed individual gene overexpression experiments for validation, and found that individual overexpression of *SEC65*, *MNS1*, *SWA2*, *ERV2*, and *ERO1* significantly increase the α-amylase production rates by different levels (1.32 to 2.2-fold) (Fig. 6b, Supplementary Data 17). Sec65 is one out of six subunits of the signal recognition particle (SRP), which is involved in protein targeting to the ER[47]. Overexpression of *SEC65* would be anticipated to increase the SRP-dependent co-translational translocation, which would benefit protein translocation from cytosol to ER. Mns1 is involved in folding and ERAD, which is responsible for the removal of one mannose residue from a glycosylated protein. α-amylase contains multiple *N*-glycosylation sites, and therefore would be benefited from *MNS1* overexpression from facilitated proper folding. *ERO1* encodes a thiol oxidase required for oxidative protein folding in the ER and provides Pdi1 with oxidizing equivalents for disulfide bond formation[2]. We observed that overexpression of *ERO1* has a positive effect on α-amylase production (2-fold). Overexpression of *ERO1* has also been shown to enhance disulfide-bonded human serum albumin (HSA) secretion in *Kluyveromyces lactis*[48] and single-chain T-cell receptors (scTCR) and single-chain antibodies (scFv) secretion in *S. cerevisiae*[49]. To be noted here, *ERO1* has also been predicted as the overexpression target for recombinant protein overproduction from a simple yeast oxidative model[50]. Therefore, *ERO1* might be considered as a generic target for secretory protein production. *SWA2* is important for vacuole sorting, here we also show that by overexpressing this gene, there is increased α-amylase production (Fig. 6b).

From four metabolic gene targets, only overexpression of *CYS4* led to a significant increase (2.14-fold) of α-amylase productivity (Fig. 6c). Cys4 (Cystathionine beta-synthase) is involved in cysteine synthesis. Comparing the amino acid composition of α-amylase with the average amino acid composition of *S. cerevisiae*, we identified that there is a 9-fold enrichment for cysteine in α-amylase compared with the general yeast proteome (Supplementary Table 1), which explains why overexpression of *CYS4* drastically increases the α-amylase production rate. Crs1 (Cysteinyl-tRNA synthetase), which is responsible for cysteinyl-tRNA aminoacylation by coupling cysteine to cysteinyl-tRNA, was also predicted as an overexpression target. However, overexpressing this gene did not significantly increase the α-amylase production rate. The other two metabolic targets are Gna1 (Glucosamine-6-phosphate acetyltransferase) and Pcm1 (Phosphoacetylglucosamine mutase), which are related to the synthesis of the *N*-glycosylation-precursor *N*-linked oligosaccharides. Overexpression of the corresponding genes did not significantly increase α-amylase production rates, suggesting that *N*-glycosylation precursor synthesis may not be the bottleneck for α-amylase production.

In total, for all chosen targets in the secretory pathway, 9/14 were validated as positive targets, while for identified metabolic

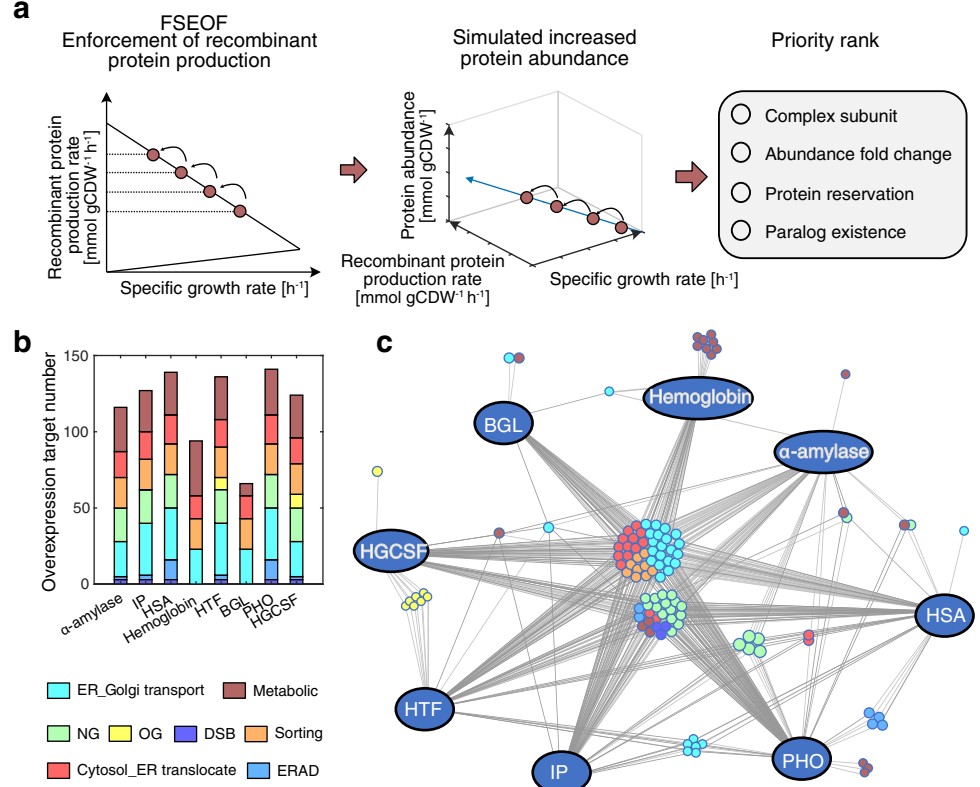

**Fig. 5 Prediction and comparison of overexpression targets for improving recombinant protein production. a** Adapted FSEOF method for target identification. Firstly, we reduced the specific growth rate in the simulation. The carbon flow towards the growth can be diverted to the recombinant protein production by maximizing the recombinant protein production. Then from the simulated native protein abundances, we can select those proteins with simulated abundance increase resulted from the enforcement of recombinant protein production as initial targets. Priority rank was then performed to further select the targets. Several parameters were considered during this rank such as whether the target is a subunit from an enzyme complex, whether the target protein abundance has changed significantly, whether the protein is abundant in the cell from the PaxDb, whether the protein contains paralogs. **b** Pathway overview of the predicted overexpression targets for eight recombinant proteins, respectively. **c** Comparison of predicted targets for the eight recombinant proteins. Only targets with high priority (higher than 3) were shown in this figure. The figure is visualized using the DiVenn[78]. Source data are provided as a Source Data file.

targets, the accuracy was 1/4. Besides the higher accuracy in the secretory targets compared with metabolic targets, FSEOF gives more targets in the secretory pathway even though the fraction of metabolic enzymes in the model is much higher. This may give us a hint that for recombinant protein secretion, the secretory pathway is more likely to be the bottleneck, and these results also demonstrate the value of the presented mathematical model for dissecting and systematic analysis of the role of complex protein secretory pathway in recombinant protein production and strain development.

## Discussion

In this study, we presented a genome-scale model of yeast that integrates metabolism, protein translation, protein post-translational-modification, ERAD and sorting processes. The model enables the calculation of unit secretory cost of any protein that is processed by the secretory pathway. We have shown that the model can correctly predict the switch from the use of high-affinity to low-affinity glucose transporter as a result of resource optimization (Fig. 2). With the unit secretory cost calculation and reported transcriptome data, we also detected that upon expression of a recombinant protein, which is processed by the secretory pathway, yeast optimizes the limited secretory capacity by down-regulating the expression of secretory proteins that are expensive to process (Supplementary Fig. 2). These two simulations suggest

that the cell allocates its limited resources by an optimization strategy, which can be accomplished through regulatory networks that have been evolved through the long history of yeast upon extracellular and intracellular environments[51,52].

We next used the model to simulate protein misfolding and retention of CPY and hereby identified that there is a certain ER tolerance to the misfolded protein (Fig. 3). Parameter sensitivity analysis showed the importance of retro-translocation in ER stress. This suggests that increasing the level of retro-translocation may alleviate the ER stress caused by the retention of misfolded protein. Since quality control and ERAD pathways are highly conserved between yeast and higher eukaryotes, this may indicate targets for treating a number of human diseases related to misfolded protein accumulation such as Alzheimer's and Parkinson's[53–55], which has been reported as therapeutic interventions[56,57]. This analysis suggests the potential of pcSecYeast to investigate the mechanism behind the misfolding fitness cost by simulating numerous hypotheses. This model is a proof of concept, and it could be further applied to study the importance of protein secretory pathway involvement in human diseases, e.g., the unfolded protein response (UPR) system in cancer cells, which is strongly activated by high accumulation of misfolded proteins in the ER[58]. Adopting pcSecYeast concept into a cancer cell line, for example, will allow to simulate and get a more systematic understanding of the UPR system overactivation in cancer cells in the future.

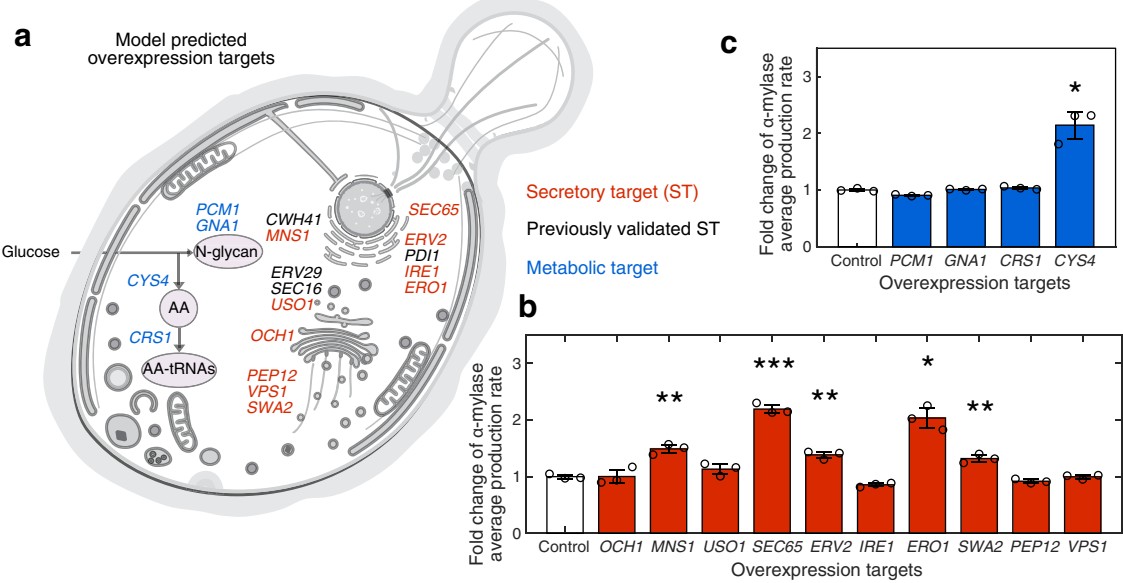

**Fig. 6 Validation of selected predicted overexpression targets for α-amylase overproduction. a** Protein localization of the selected predicted overexpression targets. Yeast compartmentalized figure is from SwissBioPics under CC BY 4.0 license. **b** Validation result of predicted secretory targets. c) Validation result of predicted metabolic targets. Statistical analysis was performed using a Student's t-test (two sample, two tailed, unequal variance, *: $P < 0.05$, **: $P < 0.01$, ***: $P < 0.001$). The gene fragments were amplified from the yeast genome and assembled into the pSP-GM1 expression vector under the control of *TEF1* promoter, respectively. Data are shown as average values ± standard errors of independent biological triplicates. α-amylase was under stable expression on the multicopy plasmid CPOTud under the control of *TPI1* promoter in a *tpi* deletion background strain. GNA1 (Glucosamine-6-phosphate acetyltransferase); PCM1 (Phosphoacetylglucosamine mutase); CRS1 (Cysteinyl-tRNA synthetase);CYS4 (Cystathionine beta-synthase); CWH41 (Processing alpha glucosidase I); OCH1 (Mannosyltransferase of the cis-Golgi apparatus); MNS1 (Alpha-1,2-mannosidase); USO1(Intracellular protein transport protein from ER to Golgi); SEC65 (Signal recognition particle subunit); ERV2 (FAD-linked sulfhydryl oxidase); IRE1 (Serine/threonine-protein kinase/endoribonuclease); ERO1 (Endoplasmic oxidoreductin-1); SWA2 (Auxilin-like clathrin uncoating factor); VPS1 (Vacuolar protein sorting-associated protein); ERV29 (ER-derived vesicles protein); PEP12 (Syntaxin); PDI1 (Protein disulfide-isomerase); SEC16 (COPII coat assembly protein). *P* values: $P^{PCM1}$: 0.0119, $P^{GNA1}$: 0.6231, $P^{CRS1}$: 0.1728, $P^{CYS4}$: 0.0199, $P^{OCH1}$: 0.9804; $P^{MNS1}$: 0.0048; $P^{USO1}$: 0.1723; $P^{SEC65}$: 0.0004; $P^{ERV2}$: 0.003; $P^{IRE1}$: 0.0119; $P^{ERO1}$: 0.0115; $P^{SWA2}$: 0.0078; $P^{PEP12}$: 0.0469; $P^{VPS1}$: 0.9431. Source data are provided as a Source Data file.

Rational design for recombinant protein production is a crucial task due to the importance of recombinant protein market share, but a very difficult task due to the complexity of the secretory pathway. pcSecYeast serves as a platform for the rational design of system-level engineering targets for recombinant protein production (Figs. 5, 6). Besides experimentally validating the predicted engineering targets for α-amylase production (Fig. 6), we further noticed consistency between predicted targets for other recombinant proteins and literature reports, such as *HEM2*, *HEM3*, and *HEM12* for hemoglobin production[59]. We confirmed that even though *HEM4* is also in the heme synthesis pathway, this is not a rate-limiting step in the heme synthesis[59]. According to the priority rank from the model prediction, Hem4 has lower predicted priority score compared with other proteins such as Hem2 and Hem3. In addition, for targets that were predicted with nonsignificant impact when overexpressed, we found that previous studies to report similar results. For example, over-expressing vacuolar sorting gene *SEC15* and *SEC4* has been shown to have no positive impact on α-amylase production[45] (Supplementary Data 9).

To be noted here, our model captures most of the secretory processes, but currently exclude some processes such as Endosome and Golgi-associated degradation pathway (EGAD)[60], the unfolded protein response and other signaling and regulatory networks[61]. Therefore, including those processes could potentially increase the prediction accuracy, in particular when it comes to the dynamic aspects of protein secretion. Besides that, we simplified some processes to perform the simulation, which would also introduce some uncertainties, for example, different types of glycans and glycoforms can exist for *N*-glycosylation[62].

However, modifications to incorporate these processes in the model will be relatively easy in case there is a need to study specific proteins where these processes are important.

In conclusion, we present pcSecYeast as the genome-scale model which allows systematic modeling of the protein secretory pathway and its interaction with metabolism and gene expression in yeast. This model enables the systematic prediction of engineering targets for recombinant protein production, from both the metabolic and secretory part of the model. The model facilitates in silico testing of various hypotheses for specific protein expression, while the predicted targets are validated to be suitable for the application. With this advancement, we expect that this type of powerful genome-scale secretory model could also be developed for other recombinant protein-producing cells, which will entail a fully in silico hypothesis generation and identification of cell engineering targets for strain development.

## Methods

**Construction of pcSecYeast and constraint-based analysis**. We reconstructed pcSecYeast, which accounts for cell metabolism and protein synthesis processes. Detailed instruction can be found in Supplementary Method 2–6 and Supplementary Figs. 9–13. The reconstruction is based on the latest yeast GEM, Yeast8.3.5[17]. Firstly, we refined all protein PTM precursors synthesis reactions in the model, such as dolichol synthesis for *N*-glycosylation, GPI anchor synthesis for GPI modification (Supplementary Data 1). Missing reactions in those precursor synthesis pathways with corresponding GPRs and necessary transport reactions were added into the model for gap-filling.

We split all reversible enzymatic reactions into forward and reverse reactions, and split reactions catalyzed by isozymes into multiple identical reactions with various isozymes to facilitate substrates and EC number annotation extraction steps in further $k_{cat}$ match process. Besides that, we formulated protein synthesis reactions for all proteins in the model. To facilitate the reconstruction process, the

protein synthesis and secretion were divided into 12 different processes: protein translation, protein translocation, ER N-glycosylation, disulfide bond formation, ER O-glycosylation, GPI anchor transfer, COPII anterograde transport, COPI retrograde transport, Golgi N-glycosylation, Golgi O-glycosylation, versatile vesicular transport to destination compartment. Compared with other fine-grained proteome constrained models, transcription was not included in pcSecYeast, as it was shown that adding transcription does not impact model predictions due to the strong linear correlation of transcription with translation[63]. While transcription was not added in the model, both the energy cost of transcription and the cellular RNA content were included in the biomass equation of pcSecYeast. Thus, adding the transcription would drastically increase the model complexity and lower the simulation efficiency without necessarily improving model predictive strength. Furthermore, translation processes such as translation initiation, elongation, and termination were lumped into one reaction since those reactions were also linearly correlated and the amount of the energy and resources used in translation was the main information to capture in the simulations. Protein-specific information matrix (PSIM) and localization information for all proteins used in further protein modification steps were downloaded from UniPort[64] and the SGD[65] database (Supplementary Data 6). We formulated these processes into 72 template reactions. Using the template reactions, we formulated protein synthesis reactions for all proteins in the model. To represent the abundance of unpresented proteins that go through ER, we added a dummy ER protein in the model which uses the same composition as the protein in the biomass protein, and the PTM for the dummy ER protein is calculated as the mean protein modification for proteins that pass through the secretory pathway using the protein abundance from PaxDb[22] and PSIM information. Protein content in the biomass was used to represent protein abundance for proteins excluded in the model. The ratio was rescaled from 1 in the original GEM Yeast8 to a lower value 0.3, which was estimated based on the fact that all proteins in the model taking up roughly 70% of the total proteome according to the PaxDb database. Detailed model construction and constraints coupling can be found in Supplementary Method 2–6. RAVEN2 toolbox[66] and COBRA toolbox[67] were used in the reconstruction.

**Model simulation for growth using glucose concentration as the constraint.** Since the specific growth rate is integrated into the coupling constraints, we adopted a binary search method when we simulated growth. For each specific growth rate, we sampled the glucose concentration until the minimal glucose concentration that can sustain the growth was found. The glucose concentration was used to calculate kinetics using the Michaelis–Menten equation where $K_M$ and maximal uptake rate $k_{cat}$ of glucose transporters were collected from the literature[68–70]. As for the glucose transporters which does not have any $k_{cat}$ values, the $V_{max}$ data was used to convert to $k_{cat}$ values with the assumption that the expression levels are comparable in the collected dataset since they expressed transporter constructs under constitutive promoters in a yeast glucose-transporter null-mutant[24,69,71]. The model was set with minimal media and the dummy protein production was set as the objective. Due to the requirement of the linear programming (LP) solver (SoPlex, https://soplex.zib.de), all constraints were written in a LP file for solving in each simulation[21,72]. This method for adding constraints is used in all following simulations unless otherwise stated.

**Estimation of unit secretory cost and direct cost for secretory proteins.** Unit secretory cost of synthesizing about 500 proteins that localize to the cell membrane or are secreted were estimated using the model. At a specific growth rate of 0.1 h$^{-1}$, we used pcSecYeast to produce a sequential small fraction production of those proteins, respectively. The glucose uptake rate minimization was set as the objective. Using the simulated glucose uptake rates and the production rates, we could fit the linear equation to get the slope which is the unit secretory cost for each protein. This cost stands for the energetic cost for synthesizing the protein, PTM, sorting and even the related cost for the corresponding fraction of the catalytic machineries in these processes.

Direct cost accounts for the energetic cost for synthesizing the amino acids, bounded glycan precursors and enzyme bounded energetic molecules, which was calculated with only the basic GEM constraints including the mass balance and reaction bound, without any enzyme-related constraint. Since this simulation does not require any extra constraint, we used the optimize function and default Gurobi solver in COBRA toolbox[67] rather than the SoPlex and LP file method.

**Estimation of secretory cost for glucose transporters.** Secretory cost specifies the cost for utilizing each glucose transporter to sustain a given glucose uptake rate and the corresponding growth rate, respectively. The secretory cost can be calculated as the required abundance of the transporter multiplied by the unit secretory cost:

$$\text{Secretory cost}_i = \text{unit secretory cost}_i \cdot \left[E_i\right] = \text{unit secretory cost}_i \cdot \frac{V_{\text{glc,total}}}{k_{\text{cat},i} \cdot \frac{[S]}{[S]+K_{\text{M},i}}}$$

(1)

**Analysis of gene expression versus protein unit secretory cost.** Absolute transcriptome data for three strains (AAC, MH34, and B184) with different

α-amylase production levels were used for the correlation analysis (Supplementary Data 18)[25]. Pearson correlation coefficient was used to assess the correlation of unit secretory costs with the expression levels.

**Simulation of protein misfolding and accumulation.** We used CPY as an example to show how the model responds toward misfolded protein production. CPY was expressed in the model with different levels from the native abundance (native expression level) towards its 25-fold levels as reported in the literature[32] by constraining its translation flux. In order to identify the factor causing the accumulation of misfolded protein in the ER, we performed the parameter sensitivity analysis for ERAD capacity, ER volume, ER membrane space, total secretory machinery capacity and retro-translocation complexes abundance, respectively. Since the membrane space and the volume of proteins are positively correlated with the protein weight[73], ER membrane space and ER volume constraints can be converted to proteome abundance constraints, which can be calculated from the proteome data. Therefore, all these parameters can be constrained by an upper limit on the total abundance of the corresponding proteins. In the meanwhile, we changed the misfolding ratio constraint of CPY by coupling the flux of misfolding reaction and the translation reaction of CPY. When misfolded protein was retained in the ER, we used the multiple rounds reactions of binding Kar2 and Pdi1 to reflect the occupancy of Kar1 and Pdi1 as reported[2,32]. The coefficient of this reaction was used to represent the time for the retention. For simulations of the combination of CPY expression levels and misfolding ratio, we used the binary search as mentioned above to search for the maximum specific growth rate. The accumulated CPY rate was obtained from the simulated flux under the maximum specific growth rate condition. To reflect the CPY production as close to the in vivo as possible, we adjusted the N-glycans attached to the N-glycosylation sites of CPY[74].

**Expansion of pcSecYeast to recombinant protein specific models.** We expanded pcSecYeast to represent the recombinant protein production by adding the production and secretion reactions using the same template reactions for the native secretory proteins. The PTMs, amino acid sequences and leader sequences were collected from the literature. Detailed information for those proteins and the literature reference can be found in Supplementary Data 7.

**Model simulation for recombinant protein production.** To simulate recombinant protein production, the model was constrained with a certain specific growth rate, and then the recombinant protein production was maximized. SD-2×SCAA medium was used in the simulations[42]. All constraints mentioned except the specific parameters used in the parameter sensitivity analysis were added when writing the LP file for solving by SoPlex (https://soplex.zib.de).

**Machine learning for protein feature importance analysis towards the protein production.** Machine learning was integrated to score the importance of factors. In this study, various factors (PTMs, amino acid compositions) were used as the input features and the maximum recombinant protein production rate was used as the target label. We split the created dataset into a training dataset and testing dataset at the ratio of 80% and 20%, respectively. A random forest regressor with 10 estimators was used to train the model. Feature importance scores from the random forest were computed by SHAP (SHapley Additive exPlanations)[75]. Python (3.7.6) with SHAP (0.39.0), scikit-learn (0.23.2), pandas (1.1.3), SciPy (1.5.2), NumPy (1.20.2) and Matplotlib (3.3.2) were used in the analysis and visualization. Five-fold cross validation was performed.

**Overexpression target prediction for recombinant protein overproduction.** Identification of overexpression targets for improving recombinant protein production was performed using the concept of FSEOF[43] but to identify the proteins with increased expression during the enforcement of recombinant protein production. To be noted here, original FSEOF searches for the candidate fluxes to be amplified through scanning for those fluxes that increase with enforced product formation flux under the objective function of maximizing biomass formation flux, which is under the assumption that there is a tradeoff between growth and target production. pcSecYeast is much more complex than the basic GEM and can better represent the cell state, which the recombinant protein production does not always increase with the decrease of growth. Besides that, there is metabolic state switch of the fermentation ratio for energy production. Therefore, to eliminate growth and metabolic state influence, we selected a small window (0.25 h$^{-1}$-0.3 h$^{-1}$) for this analysis. In this window, we reduced the growth rate in uniform small intervals and maximized the recombinant protein production rate to perform simulations. The carbon flux towards biomass production was instead diverted to recombinant protein production. As a result, the model can predict abundances for all native proteins in each simulation. From all simulations, we related the abundance changes for each native protein to the reduction in growth rate and the enforcing increase in recombinant protein production rate. The native proteins with amplified expression accompanied increased recombinant protein production were selected as initial potential overexpression targets. In order to reduce the potential target number for experimentation, we used several cut-offs to rank the priority for those predicted targets: 1) for proteins that always increase with the

enforcement of the recombinant protein production with a Spearman correlation score 1, the priority score was set to 1; 2) for proteins with priority score 1 and showed 1.2-fold abundance change of the maximum recombinant protein production state towards the maximum specific growth rate, the priority score was set to 2; 3) for proteins with priority score 2 and showed a comparable difference towards the reference PaxDb abundance, which represents the reservation state of the protein abundance in the cell, the priority score was set to 3; 4) for proteins with priority score 3 and were neither subunits of complexes nor contain paralogs, the priority score was set to 4. Proteins with the priority score close to 0 in the result indicate those proteins are not identified as overexpression targets. Targets with higher priority scores should be prioritized for overexpression. Proteins with priority score lower than 0 should be considered as downregulation targets. Based on the criteria, we ranked the targets and generated annotated tables as result for all tested eight recombinant proteins, respectively (Supplementary Data 8–15). For plotting the common targets shared by all eight recombinant proteins analyzed in this study, we only chose the priority score of 3 and 4 for the analysis. As for the predicted overexpression targets for $\alpha$-amylase overproduction, we grouped those proteins based on their functions (Supplementary Fig.8a) and selected 18 proteins, which covers most of the function and ranked with high priority score for further validation (Supplementary Fig. 8).

**Experimental validation**. All strains and plasmids used in this study are listed in Supplementary Table 2. Plasmids for gene overexpression were constructed by insertion of the gene fragment, which was amplified from the yeast genome then assembled with the expression vector pSPGM1 through Gibson assembly method. The standard LiAc/SS DNA/PEG method was used for yeast transformation.

For strain constructions, yeast strains were grown in SD-URA medium at 30 °C according to the auxotrophy of the cells. For $\alpha$-amylase production in shake flasks, yeast strains were cultured for 96 h at 200 rpm, 30 °C with an initial $OD_{600}$ of 0.05 in the SD-2×SCAA medium containing 20 g $L^{-1}$ glucose, 6.9 g $L^{-1}$ yeast nitrogen base without amino acids, 190 mg $L^{-1}$ Arg, 400 mg $L^{-1}$ Asp, 1,260 mg $L^{-1}$ Glu, 130 mg $L^{-1}$ Gly, 140 mg $L^{-1}$ His, 290 mg $L^{-1}$ Ile, 400 mg $L^{-1}$ Leu, 440 mg $L^{-1}$ Lys, 108 mg $L^{-1}$ Met, 200 mg $L^{-1}$ Phe, 220 mg $L^{-1}$ Thr, 40 mg $L^{-1}$ Trp, 52 mg $L^{-1}$ Tyr, 380 mg $L^{-1}$ Val, 1 g $L^{-1}$ BSA, 5.4 g $L^{-1}$ $Na_2HPO_4$ and 8.56 g $L^{-1}$ $NaH_2PO_4 \cdot H_2O$ (pH = 6.0)[42].

The $\alpha$-amylase activity was measured using the $\alpha$-amylase assay kit (Megazyme) with a commercial $\alpha$-amylase from *Aspergillus oryzae* (Sigma-Aldrich) as the standard. Samples were centrifuged for 10 min at 15,000 g, 4 °C and the supernatant was used for extracellular $\alpha$-amylase quantification.

**Reporting summary**. Further information on research design is available in the Nature Research Reporting Summary linked to this article.

## Data availability

Protein Specific Information Matrix (PSIM) information for all proteins in *S. cerevisiae* was collected from literature and UniProt database. Proteome and transcriptome data used in this study was collected from literature and PaxDb database. Enzyme turnover numbers ($k_{cat}$ values) were collected from BRENDA database. Simulated costs and predicted targets for recombinant protein overproduction are also provided in the Supplementary Data. All data used in this study are included in Supplementary Data and GitHub repository [https://github.com/SysBioChalmers/pcSecYeast][76]. Intermediate results are available in the Zenode [https://doi.org/10.5281/zenodo.6320643][77]. Source data are provided with this paper.

## Code availability

To facilitate further usage, we provide all codes and detailed instruction in GitHub repository [https://github.com/SysBioChalmers/pcSecYeast]. Descriptions of the code can be found in the Supplementary Method 2–6. All codes to reproduce figures were also included in the GitHub repository.

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

## Acknowledgements

We thank Olena P Ishchuk for providing the hemoglobin sequence for simulation. This project has received funding from the Novo Nordisk Foundation (grant no. NNF10CC1016517, J.N.), VINNOVA center CellNova (2017-02105, F.L.), the Knut and Alice Wallenberg Foundation (J.N.), and the European Union's Horizon 2020 research and innovation program with projects DD-DeCaF (grant no. 686070, J.N., F.L. and Y.C.). The computations were enabled by resources provided by the Swedish National Infrastructure for Computing (SNIC) at Chalmers Centre for Computational Science and Engineering (C3SE) and High Performance Computing Center North (HPC2N), partially funded by the Swedish Research Council through grant agreement no. 2018-05973 (F.L. and Y.C.).

## Author contributions

F.L. and J.N. designed the research. F.L. performed the research. Y.C. contributed to the model simulation. Q.Q. and Y.W. performed the experimental validation. L.Y. contributed to the protein feature importance analysis. I.E.E. contributed to the model reconstruction. F.L., Y.C., Q.Q., Y.W., M.H., I.E.E., A.F., E.J.K. and J.N. analyzed the data. F.L., Y.C., E.J.K. and J.N. wrote the paper. All authors approved the final paper.

## Funding

## Competing interests

The authors declare no competing interests.
