## [Peer Review File · Nature Communications]

Improving recombinant protein production by yeast through genome-scale modeling using proteome constraintsReviewers' Comments:

Reviewer #1:

Remarks to the Author:

In this manuscript, Li et al. present a computational genome-scale model of yeast *Saccharomyces cerevisiae*, which integrates its genome-scale model of metabolism with multiple other processes implicated in protein secretory pathways namely protein translation, post-translational-modification, endoplasmic-reticulum (ER)-associated protein degradation and sorting processes. This model could successfully predict the Crabtree effect and the switch from high-affinity to low-affinity glucose transporters at low vs. high specific growth rates. The authors also used the model to predict (i) how yeast responds to varying production levels of recombinant proteins by adjusting the expression of high-cost native secretory proteins, (ii) the ER tolerance to misfolded proteins and associated fitness costs, (iii) the effect of recombinant protein size and post-translational modification on their maximum production levels. They additionally used this model as a basis to rationally design promising gene overexpression targets for the enhanced production of a number of recombinant proteins. Going one step further, they also validated through literature and experimentally the predicted overexpression targets for one of the recombinant proteins.

A novel contribution of this paper is proposing a much expanded *in silico* model of secretory pathways in yeast. This work will be of interest to both systems biology community (as this modeling paradigm can be used to model secretory pathways in other organisms) as well as to the biotechnology community (since this model can streamline the rational design of microbial cell factories for recombinant protein overproduction). Overall, this is a well-written paper and an important contribution to the field of systems biology. I encourage the authors to address the following concerns to improve the quality of their manuscript:

1. The manuscript describes similarities between the proposed model and the Metabolic-Expression (ME) models or whole-cell models for yeast; however, differences between these models have not been discussed well. For example, line 92 of the manuscript says pcSecYeast accounts for protein expression and translation but all what I see in supplementary methods is protein translation, and transcriptional processes have not been accounted for in this model. On the other hand, ME models such as yETFL (PMID: 34373465), do not account for protein folding and post-translational modification processes that were included in the pcSecYeast model. Therefore, a better comparison of the scope, differences and applicability of the proposed model and the existing yeast whole-cell (PMID: 32022245 & Elseman et. al, bioRxiv, 2021) and ME models (PMID: 34373465) seems necessary.

2. Modeling protein translation:

- a. Is there any particular reason why the authors model only protein translation but not gene transcription?
- b. It is unclear whether a separate template reaction was used for translation initiation, translation elongation, and translation termination, or all these three steps were modeled using one lumped reaction.

3. It will be helpful to clarify what types of proteins are accounted for in this model. It sounds like the model accounts for metabolic enzymes and proteins involved in secretory pathways but not for other proteins (e.g., transcription factors, etc.). On a related note, how many of the 1,639 genes included in this model are metabolic genes and how many are non-metabolic genes?

4. In Equation (9) of supplementary methods, it sounds like the enzyme degradation term was neglected. Please provide justification for why this term was ignored.

5. Supplementary methods, Equation (14): What is the difference between ribosome synthesis and ribosome assembly? To my knowledge, ribosome synthesis and assembly are the same thing.

Additionally, ribosome synthesis/assembly is often considered as a non-enzymatic spontaneous reaction, so the authors should justify the use of an equation similar to Equation (12) for this reaction.

6. It is not clear from the formulation of the scSecYeast model why reversible reactions had to be broken down into forward and backward reactions (the current explanation in the paper is not enough to figure this out).

7. Were machine learning simulations to determine feature importance performed with cross validation? Nothing is mentioned in Methods about this.

Reviewer #2:

Remarks to the Author:

This study introduces a new genome scale model for studying the protein secretion machinery of the yeast *S. cerevisiae*. The model will be useful for metabolic engineering applications to improve recombinant protein production.

The major concern I have is that all the predictions in the article seem obvious. The switching of glucose transporters could be explained by enzyme binding constants. The conclusion that misfolding and retention imposes greater fitness costs than correct folding and degradation respectively is obvious again. Finally, the finding that overexpression of enzymes in secretory pathway improves the yield of protein production also seems to be trivial. No surprising or negative control predictions were made, for example of secretory proteins that need to be downregulated to improve yield. It is also not clear how 17 targets were chosen for this. A high-level overview without specific numbers is provided in the methods section. No quantitative comparison was made for these results as well, such as through correlation, precision, recall, hypergeometric tests or AUC values.

For CPY misfolding analysis, the simulated data was compared with experimental data for YFP. Why can't they directly model YFP? If CPY is well-studied (line 202), why isn't data from CPY used?

At the very end of the methods section (line 540), the authors describe the use of machine learning for feature importance. But 'machine learning' is not mentioned in the main text at all. It is unclear where and why ML was used. Isn't this model supposed to be more mechanistic than ML?

Although the authors claim this is the first secretory model, a similar model exists for CHO cells. A longer comparative explanation is needed than what is provided in the introduction.

Line 89, state number of reactions added

Line 103/104, state contribution of metabolic vs secretory proteins for both mass and total genes

Line 184, state correlation value

Reviewer #3:

Remarks to the Author:

The manuscript by Li et al. describes the construction of a genome scale metabolic model of the yeast secretory pathway (pcSecYeast) and its application to simulate the production of different misfolded or recombinant proteins. Finally, targets that increase protein production through their overexpression were predicted for 8 different recombinant protein models and experimentally verified for alpha-amylase production.

The construction of pcSecYeast represents a major and long-awaited leap forward in this highly important research area. The authors nicely demonstrate its applicability and predictive capacity for recombinant protein production in yeast, which becomes an ever more increasing field in academia and industry. The considerations raised in this manuscript will also pave the way for protein-constraint secretory models of higher eukaryotes.

The methodology is sound and meets the standards in the field.

Several of my comments are meant to improve the readability and make the manuscripts easier to understand for the broad audience of Nat Communications.

- It is more common to use "eukaryotic cells" than "eukaryal cells"
- Lines 33-34: I do not understand how "retro-translocation of misfolded proteins contributes to protein retention in the Endoplasmic reticulum (ER)"? If a protein is retro-translocated, it is not in the ER anymore, but in the cytosol.
- Lines 35-36: Please change to: ... simulate the production of various recombinant proteins and ... for overproduction of different recombinant proteins.
- Line 43: For yeast, about 500 proteins are predicted to enter the secretory pathway (see e.g. Delic et al, reference 39), which would account to approx. 10%
- Line 49: the sentence starting with "The unique modification profile ..." is difficult to read, please modify.
- Fig 1A/Text in lines 94ff: Please add how many reactions were added, and whether they are purely stoichiometric, or whether they account for repeated interaction with client proteins during folding.
- Fig 1B: I am astonished that so many proteins are involved in translocation. Does this also include folding chaperones and quality control?
- Lines 138ff: Difficult to read, please split the sentence starting with "Thus, at low specific growth rates..."
- Figure 2 and related text: I agree that it is possible to calculate the "secretory cost" of a transporter, but I do not see any evidence that this is truly the driving force that determines which transporter is used. The hypothesis should be proven with experimental data that indeed the Hxt7 transporter is less abundant at low glucose than Hxt1/3 at high glucose, or that overexpression of Hxt7 is more stressful for the cells than overexpression of Hxt1/3 (either by the authors or by literature data).
- Fig 2B and C: Currently the secretory cost or protein abundance seems to be inferred from the qGlc. It remains unclear to me why the secretory costs change with glucose concentration, the "unit costs" should be independent of such an external condition. Also, why are costs of Hxt1/3 higher at low glucose?
- Looking at the data in Table S3, the results in Figure 2C even get less comprehensible to me: the direct costs and unit costs of Hxt7 and Hxt3 are almost equal (834 vs 836 mol glucose/mol protein direct costs; 3347 vs 3349 mol glucose/mol protein unit costs), also for Hxt1 there is not a huge difference (786 and 3261 mol/mol, respectively). Please explain in more detail how this can lead to the differences observed in Fig 2B, and check carefully if a circular argument might have been generated, through the calculation of protein abundance via its catalytic properties in Eq 1 rather than using protein abundance from proteomic resources.
- Please add the systematic names of all genes/proteins that are discussed in the text (or add the short names to the supplementary tables S3 and S4).
- The text in lines 125-161 is rather hard to read, sometimes sentences are connected without a connecting word, sometimes words are missing or in the wrong grammatical form.
- The results described and discussed in Lines 164-193 are highly interesting, even though not unexpected. It might be interesting to look a bit more in detail which are the most costly secretory proteins (e.g. if there are some characteristic features or PTMs associated with being costly).
- Supplementary Figure 2b: It might be more convincing to show the correlation of one of the two higher producers rather than of AAC1.
- Why didn't the authors do the correlation to the proteome data they published previously (Qi et al. 2020. mBio 11(6):e02743-20)? Is this because they wanted to show that also in yeast transcriptional repression of secretory proteins is occurring as described for mammalian cells and filamentous fungi (Pakula et al. 2003. J Biol Chem. 278(45):45011-20)?
- Line 199: It would be good to include at least one yeast reference here (e.g. instead of ref 24 dealing with ER stress in the heart). The recent review by Ninagawa et al. 2021 Biochim Biophys Acta Gen Subj. 1865(3):129812 might be a good option covering both yeast and mammalian systems.
- Regarding protein degradation, the authors might find this interesting: A Systematic Protein

Turnover Map for Decoding Protein Degradation. Christiano et al. 2020. Cell Rep. 33(6):108378.

- Line 212: I think it should be "...in the ER for different times"
- Line 217: ...expression of wild type YFP in the cytosol... ... was observed when expressing ...
- Line 233: Please rephrase, there seems to be something (at least a word) missing, which makes the meaning of the sentence unclear.
- Line 237ff/Supplementary Figure S4: Which reaction catalyzes this flux? To my knowledge GSSG export from the ER is not characterized. However, the predicted flux would fit to the experimental data from Merksamer et al. 2008 Cell. 135(5):933-47.
- Supplementary Figure S3: Actually, I assume that the increased abundance of Kar2 and Pdi1 in the "misfold-retention" simulations might represent UPR induction. Was activation of this stress response pathway included in the model?
- Figure 3a: There should be reversible arrows for the retention part or a connection to ERAD. I understand that might not be easy or useful to include this in the model, but in vivo ER retention is usually not the final fate in yeast, the majority of misfolded proteins will be eventually degraded, but different misfolded proteins might be retained for different times or accumulate to different abundances, leading to different stress levels as simulated.
- Line 244 ff: See also my comment below (model construction), Sec61 seems not to be part of the retro-translocon.
- Line 253: ...in the ER.
- Line 269: None of the references 29-31 seems to deal with *Pichia pastoris*, please add (e.g. review by Looser et al. 2015. Biotechnol Adv. 33(6 Pt 2):1177-93) or rephrase.
- Figure 6: please add to the legend which promoter/vector was used for overexpression and how many clones per construct were analysed.
- Line 352: Overexpression of Ero1 has recently also been predicted to be beneficial for protein production by a very limited model of oxidative folding by Beal et al. 2019. Antioxid Redox Signal. 31(4):261-274.
- Line 371: ...is much higher...
- Lines 389-390: I don't think "evolutionary" is the right word here, please rephrase the last part of the sentence.
- Line 399: I don't think "recently" fits when something has been postulated more than 10 years ago.
- Line 428: Please replace "identify" with "predict"
- Line 428: This work definitely is the first comprehensive genome scale model integrating the secretory processes, and thus also the first one to predict engineering targets in these processes. However, engineering targets for recombinant protein production have been predicted by GSMMs before, albeit not for secretory functions. I recommend to rephrase this final paragraph accordingly, and give credits to these earlier works (either here or in the introduction), that improved protein production by model predictions in various host organisms.

Model construction:

- Signal peptidase should also be involved in post-translational translocation (reading further I see it is introduced as a separate topic later, but it might be good to restructure this in the text)
- Why are the Golgi enzymes not considered in N-glycosylation? (reading further I see it is introduced as a separate topic later, but it might be good to restructure this in the text)
- Literature suggests that the Sec61 translocon is not involved in ERAD. Instead Hrd1/Der1 were found to form the channel for retrotranslocation (e.g. Wu et al. 2020 Science. 368(6489):eaaz2449.)
- Doa10 seems to be the ubiquitin ligase also for ERAD-M, not only ERAD-C (Schmidt et al. 2020. Elife. 9:e56945.).
-

- Whole manuscript and supplements: "detailly" is not recognized by any of the authoritative English dictionaries. The closest recognized term is in detail.
- The supplement detailing the model construction should be proof-read, there are several typos

including one in the model itself (tanslocate instead of translocate)

- Whole manuscript and supplements: The quality of English is varying between different passages. This needs to be amended, either by one of the authors or a native speaker.

Reviewer #4:

Remarks to the Author:

The knowledge developed in the area of heterologous protein expression in recent years has been vast and has revealed the need to apply rational protein design to achieve better results.

The authors propose a novel method to evaluate the viability of protein expression in a widely used and highly useful system.

The methodology used for the design of the experiments was adequate, and is mostly available to potential users. The authors explain it simply, personally I do not think that the objective of the article is to make the reproduction of the achieved design feasible.

However, I hope that this type of analysis will be available for the use of the scientific community soon, since it means a great advance in the area.

The conclusions reached by the working group are valid and reflect the deep analysis of the results obtained, adding value to the work carried out.

REVIEWER COMMENTS

Reviewer #1 (Remarks to the Author):

In this manuscript, Li et al. present a computational genome-scale model of yeast *Saccharomyces cerevisiae*, which integrates its genome-scale model of metabolism with multiple other processes implicated in protein secretory pathways namely protein translation, post-translational-modification, endoplasmic-reticulum (ER)-associated protein degradation and sorting processes. This model could successfully predict the Crabtree effect and the switch from high-affinity to low-affinity glucose transporters at low vs. high specific growth rates. The authors also used the model to predict (i) how yeast responds to varying production levels of recombinant proteins by adjusting the expression of high-cost native secretory proteins, (ii) the ER tolerance to misfolded proteins and associated fitness costs, (iii) the effect of recombinant protein size and post-translational modification on their maximum production levels. They additionally used this model as a basis to rationally design promising gene overexpression targets for the enhanced production of a number of recombinant proteins. Going one step further, they also validated through literature and experimentally the predicted overexpression targets for one of the recombinant proteins.

Response: We thank the reviewer for the summary and highlighting the importance of our work.

A novel contribution of this paper is proposing a much expanded in silico model of secretory pathways in yeast. This work will be of interest to both systems biology community (as this modeling paradigm can be used to model secretory pathways in other organisms) as well as to the biotechnology community (since this model can streamline the rational design of microbial cell factories for recombinant protein overproduction). Overall, this is a well-written paper and an important contribution to the field of systems biology. I encourage the authors to address the following concerns to improve the quality of their manuscript:

Response: We thank the reviewer for these nice comments. We have refined the manuscript based on these comments. All revised contents are marked in blue in the main text.

1. The manuscript describes similarities between the proposed model and the Metabolic-Expression (ME) models or whole-cell models for yeast; however, differences between these models have not been discussed well. For example, line 92 of the manuscript says pcSecYeast accounts for protein expression and translation but all what I see in supplementary methods is protein translation, and transcriptional processes have not been accounted for in this model. On the other hand, ME models such as yETFL (PMID: 34373465), do not account for protein folding and post-translational modification processes that were included in the pcSecYeast model. Therefore, a better comparison of the scope, differences and applicability of the proposed model and the existing yeast whole-cell (PMID: 32022245 & Elsemman et. al, bioRxiv, 2021) and ME models (PMID: 34373465) seems necessary.

Response: Thanks for the suggestion. pcSecYeast adopts the same fine-grained proteome constrained concept as in the proteome constrained model (pcGEM) pcYeast¹ (the model from Elsemman et. al) and Metabolism-expression model (ME model) yETFL², which couples the protein synthesis with metabolism through the enzyme kinetic capacity. The main difference of pcSecYeast with pcYeast¹ and yETFL² is that pcSecYeast has well-constructed processes for protein synthesis, folding, misfolding and degradation, which are either lumped or just omitted in pcYeast¹ and yETFL². Therefore, pcSecYeast is more like an expanded version and allows the precise analysis of cellular behaviors for different physiological conditions especially for the simulation of recombinant protein production.

The main difference of pcSecYeast with the whole cell model of yeast (WM_S288C³) is the process coverage. WM_S288C decomposes cell functionality into 26 cellular processes, while pcSecYeast covers

six of those processes including metabolism, protein translation, protein folding, protein decay, protein modification and ribosome assembly. However, the protein folding and protein decay in pcSecYeast are more comprehensive and protein-specific compared with those in WM_S288C. As for the protein modification processes, glycosylation, disulfide bond and GPI considered in pcSecYeast are not considered in WM_S288C. Moreover, fine-grained proteome constrained models (including pcYeast¹, yETFL² and pcSecYeast) are constraint-based optimization frameworks using a steady state assumption, while the whole cell model is a dynamic model which uses ordinary differential equations. Compared with the vast environment- and condition-dependent parameters requirement as in the WM_S288C, less parameters are required in pcSecYeast, which enables a more efficient simulation of cell behavior.

Action: We described the differences among those models and summarized as a table (Table R1 below) in Supplementary Methods and referred to this in main text (line 105-107). In order to clarify the pcSecYeast reconstruction process, we added three flow charts (Fig. 1-2, 5 in Supplementary Methods).

Table R1. Comparison of pcSecYeast with other models.

Models	ihGlycopastoris ⁴	Mammalian secretory model ⁵	pcYeast ¹	yETFL ²	pcSecYeast	WM_S288C ³	
Organism	P. pastoris	Mammalian cells	S. cerevisiae	S. cerevisiae	S. cerevisiae	S. cerevisiae	
Model type	Basic GEM	Basic GEM	Fine-grained proteome-constrained GEM	Fine-grained proteome-constrained GEM	Fine-grained proteome-constrained GEM	whole-cell model	
Model assumption	Steady state	Steady state	Steady state	Steady state	Steady state	Dynamic	
Constraint	Mass balance	Mass balance	Mass balance, kinetic constraint	Mass balance, kinetic constraint, thermodynamic constraints	Mass balance, kinetic constraint	-	
Processes	Metabolism	Yes	Yes	Yes	Yes	Yes	
	Transcription	No	No	No	Yes, lumped	No	
	RNA cleavage	No	No	No	No	No	
	tRNA modification	No	No	Yes	No	No	
	tRNA charging	No	No	Yes	Yes	Yes	
	Translation	For the recombinant protein, NOT for native proteins	For the recombinant protein, NOT for native proteins	Yes	Yes	Yes	Yes
	Folding	For the recombinant protein, NOT for native proteins	For the recombinant protein, NOT for native proteins	Lumped with protein-	Lumped	Comprehensive, protein-specific	Lumped

		native proteins	for native proteins	specific chaperones			
	Misfolding	No	For the recombinant protein, NOT for native proteins	Lumped	Lumped	Comprehensive, protein-specific	Lumped
	Protein translational modification	N -glycosylation	Glycosylation and disulfide bond formation	No	No	Glycosylation, GPI anchor and disulfide bond formation	Phosphorylation, acetylation and ubiquitination
	Degradation	For the recombinant protein, NOT for native proteins	For the recombinant protein, NOT for native proteins	Lumped	Lumped	Comprehensive, protein-specific	Lumped
	Sorting	For the recombinant protein, NOT for native proteins	For the recombinant protein, NOT for native proteins	Lumped	No	Comprehensive, protein-specific	Lumped
	Ribosome assembly	No	No	Yes	Yes	Yes	Yes
	Protein complex formation	No	No	Yes	Yes	Yes	Yes
	Other cell processes	No	No	No	No	No	Yes
Model application	Simulate proteome changes	No	No	Yes	Yes	Yes	Yes
	Integrate proteome data	No	No	Yes	Yes	Yes	No
	Simulate protein misfolding	No	No	No	No	Yes	Yes
	Native protein competition with recombinant protein	No	No	N/A	N/A	Yes	N/A
	Simulate engineering	Only targets in metabolic	Only targets in metabolic	N/A	N/A	Targets both in secretory and	N/A

	targets for improving recombinant targets	pathway	pathway			metabolic pathways	
--	--	---------	---------	--	--	-----------------------	--

N/A means that the description is not applicable for the specific model.

2. Modeling protein translation:

a. Is there any particular reason why the authors model only protein translation but not gene transcription?

Response: We did not include the transcription based on reasons below:

- 1) It was shown that adding transcription does not impact the fine-grained proteome constrained model prediction results, since transcription and translation fluxes have strong linear correlation⁶.
- 2) Transcription in eukaryotes is complex and it involves the complex transcriptional regulatory networks. Adding this process would drastically increase the model complexity and lower the simulation efficiency, but not necessarily improving model predictive strength.
- 3) Most of the machineries involved in transcription processes are not processed in the secretory pathway, which means adding those would not have large impacts towards secretory simulations of protein secretion in our study.
- 4) Even though the transcription is not added in the model, both the energy cost of transcription and the cellular RNA content are included in the biomass equation.

Based on the reasons mentioned above, we did not include transcription in the pcSecYeast, but if the objective of a model is to do simulation towards the transcription machinery, the users of the model can customize the model for those reactions.

b. It is unclear whether a separate template reaction was used for translation initiation, translation elongation, and translation termination, or all these three steps were modeled using one lumped reaction.

Response: All these three steps were lumped into one reaction since those reactions are also linearly correlated and the amount of the energy and resources used in the translation was the main information to capture for the simulations. Lumping them together would not have impact on the simulation result but can improve the simulation efficiency, unless if the objective is to capture the dynamics of the translation process, which is out of the scope for our model.

Action: We have revised the Supplementary Methods (line 143-144 in the Supplementary Methods) to clarify this.

3. It will be helpful to clarify what types of proteins are accounted for in this model. It sounds like the model accounts for metabolic enzymes and proteins involved in secretory pathways but not for other proteins (e.g., transcription factors, etc.). On a related note, how many of the 1,639 genes included in this model are metabolic genes and how many are non-metabolic genes?

Response: pcSecYeast includes metabolic proteins and machinery proteins involved in the secretory pathway, ribosome synthesis and protein degradation. There are in total 1,156 metabolic proteins and 483 non-metabolic proteins (ribosome, ribosome assembly factors, proteasome and secretory machinery proteins).

Action: We have updated the Supplementary Table 2 to denote the function of those 1,639 genes.

4. In Equation (9) of supplementary methods, it sounds like the enzyme degradation term was neglected. Please provide justification for why this term was ignored.

Response: We divided each protein to be either correctly folded or misfolded. The correctly folded fraction then continues to form enzyme complexes while the misfolded fraction is degraded. Even though that correctly folded protein can also be degraded *in vivo*, but most of them are firstly changed to misfolded or unfolded state caused by aging or denaturing event before degradation. Besides that, there are not such measured fractions for degradation of misfolded proteins and of correctly folded proteins. Thus, we assign all degradation parts to misfolded proteins (misfolding ratio = measured kdeg ratio). Thus, in the Eq. 9 (current Eq. 10 in the revised Supplementary Methods) for enzymatic complexes, the degradation for the complex is neglected in the model (Fig. R1).

Fig. R1 Flow chart for the protein synthesis, degradation, dilution and complex formation in the pcSecYeast. Black lines: peptide flow and synthesis. Red lines: catalytic enzymes and constraints.

Action: We have added this figure in the Supplementary Methods to facilitate understanding.

5. Supplementary methods, Equation (14): What is the difference between ribosome synthesis and ribosome assembly? To my knowledge, ribosome synthesis and assembly are the same thing. Additionally, ribosome synthesis/assembly is often considered as a non-enzymatic spontaneous reaction, so the authors should justify the use of an equation similar to Equation (12) for this reaction.

Response: We apologize for the misunderstanding. Ribosome assembly factors make new ribosomes⁷, while ribosomes catalyze protein translation (Fig. R1). Therefore, both ribosome and ribosome assembly factors need to be synthesized in the model. Ribosome synthesis is constrained by the ribosome assembly factors abundance. Ribosome and Ribosomes assembly factor composition were compiled from these references^{7,8}.

Action: We have revised Eq. 14 (current Eq. 15 in the revised Supplementary methods) to denote that $V_{syn,ribo_assembly_factor}$ is the synthesis rate of ribosome assembly factors.

$$V_{ribosome} \leq \frac{k_{cat,ribo_assembly}}{\mu} \cdot V_{syn,ribo_assembly_factor}$$

6. It is not clear from the formulation of the scSecYeast model why reversible reactions had to be broken down into forward and backward reactions (the current explanation in the paper is not enough to figure this out).

Response: We use the substrates and enzyme EC numbers of enzymatic reactions to retrieve k_{cat} values in BRENDA database. As for two directions of reversible reactions, substrates are different (products in the forward direction are substrates in the reverse reaction). Therefore, to facilitate the k_{cat} matching process, we break down the reversible reactions into forward and reverse reactions. It is a common approach done in proteome-constrained model reconstruction when k_{cat} matching is required⁹⁻¹¹.

Action: We have updated the main text (458-460) to further explain this step.

7. Were machine learning simulations to determine feature importance performed with cross validation? Nothing is mentioned in Methods about this.

Response: Yes. We have performed 5-fold cross validations. The result remained similar with Fig. 4c that post-translational modifications contribute more compared with amino acid composition.

Fig. R2 Five-fold cross validation of the feature importance analysis.

Action: We have updated the legend of Fig. 4c in the main text and added this figure as the Supplementary Figure 7.

Reviewer #2 (Remarks to the Author):

This study introduces a new genome scale model for studying the protein secretion machinery of the yeast *S. cerevisiae*. The model will be useful for metabolic engineering applications to improve recombinant protein production

Response: Thanks for the summary and stating the application of our model. All revised contents are marked in blue in the main text.

The major concern I have is that all the predictions in the article seem obvious. The switching of glucose transporters could be explained by enzyme binding constants.

Response: The switch of transporters has been seen in multiple organisms¹²⁻¹⁴. The benefit of utilizing high affinity transporters during nutrient depletion or limited conditions seems evident, but there remains question why cell would switch to low affinity transporters. There are several hypotheses for the switch including: 1) the enzyme binding constants as the reviewer mentioned, which is based on the rate-affinity trade-offs among different conditions¹⁵. Cells would choose high k_{cat} but low affinity transporters when substrates are abundant; 2) utilization of the low affinity transporter allows cell to sense the substrate depletion accurately and quickly, which would give cell more time to prepare for the substrate depletion condition¹⁶; and 3) utilization of the low-affinity transporters lowers the efflux of substrates, which contributes higher net import flux of the substrate¹⁷. Each of these hypotheses cannot perfectly explain the switch to the low affinity transporters. The discussion of these hypotheses can be found in reference¹⁷, in which they also mentioned that it is hard to perform the experimental evaluation for those three hypotheses.

pcSecYeast correctly simulated the switch of glucose transporters. The model-calculated costs also show the same switch, meaning that the costs could be a new explanation i.e., the secretory cost differs across those transporters., which is not conflicting with other hypotheses. From Eq. 1 in the main text, we can see that the trade-off hypothesis (enzyme kinetics) is included in the calculation for secretory cost. Note that the costs can only be calculated by the model. This validates the predictive power of our model in energy cost quantification of utilization different glucose transporters. We also tested even if the enzyme kinetics is the same for the high affinity glucose transporter Hxt7 and low affinity transporter Hxt1, then the Hxt1 is chosen at maximum growth rate, which suggests the importance of protein secretory energy cost besides the enzyme kinetics.

Although some prediction might seem obvious, however it is crucial to note that we have used the objectives such as switching of glucose transporters as an example to prove the functionality of the model prediction. It is estimated that there are over 400 membrane protein in the yeast cell membrane for which we have included their production in the model as well. When a glucose transporter is passing through the secretory pathway towards the cell membrane it has to compete with other cell membrane proteins for protein secretory costs. This is the main reason why only the kinetic parameters of a membrane transporter or enzyme complex are not predictive for their production in the cell. It is only by using genome scale models like our model that we can calculate the tradeoff between the production cost and preferable kinetic parameters.

Action: We rephrased the secretory cost calculation and results in the main text to clarify the importance (line 140-175).

The conclusion that misfolding and retention imposes greater fitness costs than correct folding and degradation respectively is obvious again.

Response: We agree on this, but please note that pcSecYeast is the first genome-scale model that can correctly simulate this phenotype, which suggests the potential of our model in utilization in studying the mechanism behind the misfolding fitness cost and computational testing of numerous hypotheses. As we mentioned above, our model is a proof of concept, and the model can be used to study things that are not that obvious at first glance. For example, it is known that in cancer cells UPR system is strongly activated due to the high accumulation of misfolded proteins in the ER. Adopting this model into a cancer cell line, for example, will allow to simulate and get a more systematic understanding of the UPR system overactivation in cancer cells.

Finally, the finding that overexpression of enzymes in secretory pathway improves the yield of protein production also seems to be trivial. No surprising or negative control predictions were made, for example of secretory proteins that need to be downregulated to improve yield.

Response: Thank you for this interesting comment. Before we reply to this comment in details, it is important to note that predicting overexpression target using genome-scale models has been always challenging and most of the algorithms in the GEMs community developed to predict targets to downregulate or eliminate. The reason is that predicting the outcome of in-silico knock-out simulation can be done by just removing the corresponding reaction from the model. However, evaluating impact of overexpression is challenging in cell and that is why only few algorithms have been developed to predict overexpression targets. In fact, the secretory pathway has ~500 components and the critical question is that for each specific recombinant proteins overexpression of which of these components can work. If one wants to find the answer to this question experimentally, it requires a very complicated and expensive experimental design (which we have done it previously in our lab¹⁸). However, with our model we propose an in-silico approach to lower the number of targets to a handful of targets that are predicted to be specific for each recombinant protein that can then be experimentally evaluated. It is also important to note that our model predicts both over-expression and down-expression targets to improve the protein production not only for proteins engaged in the protein secretory pathway but also among metabolic pathways. Concerning the model predicted overexpression targets here, we found several novel targets which can improve α -amylase production. To be noted here, to validate the application of predicted targets, we chose a relatively high-yield α -amylase producing strain with a clear genetic background as the control strain, in which α -amylase is stably expressed on a multi-copy plasmid CPOTud under the control of *TPI1* promoter in a *tpi* deletion background strain. The yield of our best strain B184 is 4-fold higher than for our control strain, which is achieved after several rounds of UV mutagenesis and microfluidic screening from this control strain¹⁸. From our validation result, we can see that by only overexpressing one predicted target, the α -amylase production can be improved around 2-fold, which validates the effectiveness of our model in identifying important targets. The reason why we did not perform overexpression experiment on the best strain B184 is that this strain has collected numerous mutations after several rounds of mutagenesis, which may deviate from our modeled strain background (such as enzyme kinetics or energy cost related to growth) and thus may not be a good platform for validation.

In our manuscript, we focused on discussing and validating overexpression targets. This is due to that identifying overexpression targets was thought to be more useful in terms of improving the yield of protein production, since the major obstacle for secretory protein overproduction is limited secretory capacity. However, our method is able to identify downregulation targets or negative overexpression targets (those proteins are negatively correlated with recombinant protein overproduction in the simulation would be downregulation targets). In the revised version, we have added all identified downregulation targets in the Supplementary Data 2 together with the overexpression targets (In the dataset, overexpression targets are marked with positive priority score, while downregulation targets are marked with negative priority score).

As for the negative control, our model predicted that overexpression of Sec15 and Sec4 would not improve the α -amylase yield (the priority scores are close to 0 in the Supplementary Data 2), which is consistent with previous experimental report¹⁹. As for gene downregulation, we identified the consistent downregulation in the central carbon metabolism, TCA cycle and respiration as in the experimental observation for the α -amylase production²⁰, which could validate the high quality of our model. All those gene targets are now listed in the Supplementary Data 2.

Action: We have illustrated how we performed the validation in the legend of Fig. 6. We have modified the code and output the list of both over-expression and down-expression targets in the Supplementary Data 2 for all tested recombinant proteins.

It is also not clear how 17 targets were chosen for this. A high-level overview without specific numbers is provided in the methods section. No quantitative comparison was made for these results as well, such as through correlation, precision, recall, hypergeometric tests or AUC values.

Response: We apologize that we did not explain this well. As for α -amylase overexpression targets, we ranked them based on several criteria, such as whether there exists isoenzymes for this protein, whether it is in a multi-subunits complex, whether the protein is sufficient *in vivo* (calculated from the simulated protein abundance and the reference proteome abundance of PaxDb²¹) and whether it is correlated to growth rate. The above-mentioned scenarios may hinder the overexpression performance, which are therefore ranked with low priority scores. After the rank, we clustered the proteins based on their functions and selected the 18 targets with different functions and ranked with high priority scores for validation.

Since our model is a genome-scale model, it contains 1,639 genes. We have predicted the impact of all those genes in terms of overexpression towards improving recombinant protein production. More than 500 proteins were identified from the prediction (including positive overexpression targets and many more negative overexpression targets). It would be difficult for us to perform a validation on this scale and would be a waste of resource to validate those negative targets. In our opinion the experimental results shown in the manuscript and consistent results with previous reports should be sufficient to prove the quality of our model.

Since we did not perform systematic experiments to validate all predicted results, it is not possible for us to calculate the correlation, precision, or other parameters. We do agree with the reviewer that if the systematic overexpression experiments could be done in a high throughput way in the future, this would be an excellent dataset for us to further validate our model.

Action: We added several sentences in the methods section (line 591-593) to describe how we chose these targets for validation.

For CPY misfolding analysis, the simulated data was compared with experimental data for YFP. Why can't they directly model YFP? If CPY is well-studied (line 202), why isn't data from CPY used?

Response: CPY and its derivative misfolded form CPY* are widely used to study protein misfolding^{22,23}, but the quantitative fitness cost as measured for YFP is not available for CPY. We raise the YFP data more like a control to indicate that our simulation is in the reasonable range. This renders the knowledge gained from the model simulation as more convincing, such as the retro-translocation constraint we identified for the accumulation of misfolded proteins.

Our model can be used to simulate the YFP expression, but we did not simulate YFP as this protein is a cytoplasmic protein, which is synthesized in the cytosol and not processed through the secretory pathway. Here our focus is on the protein secretion and proteins that are processed in the secretory pathway, and we are interested in studying the misfolding and accumulation of this class of proteins. As we discussed earlier, cell disorder and many human diseases are related to misfolding of proteins in ER²⁴⁻²⁶.

Action: We have removed the lines of mentioning YFP to reduce the misunderstanding. We compared our simulation with the study for expressing different levels of CPY and its misfolded form CPY* qualitatively[23] (line 224 -264 in the revised main text).

At the very end of the methods section (line 540), the authors describe the use of machine learning for feature importance. But 'machine learning' is not mentioned in the main text at all. It is unclear where and why ML was used. Isnt this model supposed to be more mechanistic than ML?
Response: We apologize that we did not explain this well. Secretory proteins are complex which involve different features such as amino acid composition, different post-translational modifications, and localization. Therefore, to tell which feature impacts the protein production most (positive and negative contribution), we used machine learning to perform a feature importance rank. The machine learning model here is just used to analyze results from our pcSecYeast model.

Action: We have now referred the method in the main text (line 290-291) and rephrased the title of the section in the methods (line 553 in the revised main text).

Although the authors claim this is the first secretory model, a similar model exists for CHO cells. A longer comparative explanation is needed than what is provided in the introduction.
Response: Our model is the first proteome-constrained genome-scale protein secretory model, rather the first secretory model. We have made a table (Table R2 below) to compare the pcSecYeast with previous secretory models including secretory models for *P. pastoris* and mammalian cells. In terms of model similarity, our model is much closer to non-secretory proteome-constrained model pcYeast[1] and ME model yETFL[2].

Compared with mammalian secretory model, our model covers more process and includes protein synthesis reactions for all proteins in the model rather than just for the recombinant protein. Therefore, our model improves the model prediction by considering the competition of native secretory proteins with recombinant protein production. Besides that, rather than a basic GEM extension as the mammalian secretory model is, our model adopts the fine-grained proteome constrained concept[27], which allows systematic modeling of the protein synthesis and the interaction with metabolism.

Action: We have included the comparison in the Supplementary Methods and referred that in the main text (line 105-107).

Table R2. Comparison of pcSecYeast with other models.

| **Models** | **ihGlycopastoris[4]** | **Mammalian secretory model[5]** | **pcYeast[1]** | **yETFL[2]** | **pcSecYeast** | **WM_S288C[3]** |
|---|---|---|---|---|---|---|
| Organism | *P. pastoris* | Mammalian cells | *S. cerevisiae* | *S. cerevisiae* | *S. cerevisiae* | *S. cerevisiae* |
| Model type | Basic GEM | Basic GEM | Fine-grained proteome-constrained GEM | Fine-grained proteome-constrained GEM | Fine-grained proteome-constrained GEM | whole-cell model |
| Model assumption | Steady state | Steady state | Steady state | Steady state | Steady state | Dynamic |

Constraint		Mass balance	Mass balance	Mass balance, kinetic constraint	Mass balance, kinetic constraint, thermodynamic constraints	Mass balance, kinetic constraint	-
Processes	Metabolism	Yes	Yes	Yes	Yes	Yes	Yes
	Transcription	No	No	No	Yes, lumped	No	Yes
	RNA cleavage	No	No	No	No	No	Yes
	tRNA modification	No	No	Yes	No	No	Yes
	tRNA charging	No	No	Yes	Yes	Yes	Yes
	Translation	For the recombinant protein, NOT for native proteins	For the recombinant protein, NOT for native proteins	Yes	Yes	Yes	Yes
	Folding	For the recombinant protein, NOT for native proteins	For the recombinant protein, NOT for native proteins	Lumped with protein-specific chaperones	Lumped	Comprehensive, protein-specific	Lumped
	Misfolding	No	For the recombinant protein, NOT for native proteins	Lumped	Lumped	Comprehensive, protein-specific	Lumped
	Protein translational modification	N -glycosylation	Glycosylation and disulfide bond formation	No	No	Glycosylation, GPI anchor and disulfide bond formation	Phosphorylation, acetylation and ubiquitination
	Degradation	For the recombinant protein, NOT for native proteins	For the recombinant protein, NOT for native proteins	Lumped	Lumped	Comprehensive, protein-specific	Lumped
	Sorting	For the recombinant protein, NOT for native proteins	For the recombinant protein, NOT for native proteins	Lumped	No	Comprehensive, protein-specific	Lumped
	Ribosome assembly	No	No	Yes	Yes	Yes	Yes
Protein	No	No	Yes	Yes	Yes	Yes	

	complex formation						
	Other cell processes	No	No	No	No	No	Yes
Model application	Simulate proteome changes	No	No	Yes	Yes	Yes	Yes
	Integrate proteome data	No	No	Yes	Yes	Yes	No
	Simulate protein misfolding	No	No	No	No	Yes	Yes
	Native protein competition with recombinant protein	No	No	N/A	N/A	Yes	N/A
	Simulate engineering targets for improving recombinant targets	Only targets in metabolic pathway	Only targets in metabolic pathway	N/A	N/A	Targets both in secretory and metabolic pathways	N/A

N/A means that the description is not applicable for the specific model.

Line 89, state number of reactions added

Action: We have added the number in line 94 in the revised main text.

Line103/104, state contribution of metabolic vs secretory proteins for both mass and total genes

Action: We have listed the contribution in the line 111-114.

Line 184, state correlation value

Action: We have added the correlation value in the main text (line 199).

Reviewer #3 (Remarks to the Author):

The manuscript by Li et al. describes the construction of a genome scale metabolic model of the yeast secretory pathway (pcSecYeast) and its application to simulate the production of different misfolded or recombinant proteins. Finally, targets that increase protein production through their overexpression were predicted for 8 different recombinant protein models and experimentally verified for alpha-amylase production.

The construction of pcSecYeast represents a major and long-awaited leap forward in this highly important research area. The authors nicely demonstrate its applicability and predictive capacity for recombinant protein production in yeast, which becomes an ever more increasing field in academia and

industry. The considerations raised in this manuscript will also pave the way for protein-constraint secretory models of higher eukaryotes.

The methodology is sound and meets the standards in the field.

Response: Thanks for this kind summary of our work and highlighting the importance of our work. We really appreciate it.

Several of my comments are meant to improve the readability and make the manuscripts easier to understand for the broad audience of Nat Communications.

Response: Thanks for these nice comments! All revised contents are marked in blue in the main text.

- It is more common to use “eukaryotic cells” than “eukaryal cells”

Response: We have fixed this in the revised main text.

- Lines 33-34: I do not understand how “retro-translocation of misfolded proteins contributes to protein retention in the Endoplasmic reticulum (ER)”? If a protein is retro-translocated, it is not in the ER anymore, but in the cytosol.

Response: Thanks for this comment. It should be that the limited retro-translocation *capacity* precludes the degradation of excessive misfolded proteins, which would lead to the protein retention of misfolded proteins.

Action: We revised the lines (33-35) in the main text.

- Lines 35-36: Please change to: ... simulate the production of various recombinant proteins and ... for overproduction of different recombinant proteins.

Action: We have revised the main text (line 35-36).

- Line 43: For yeast, about 500 proteins are predicted to enter the secretory pathway (see e.g. Delic et al, reference 39), which would account to approx. 10%

Action: We have revised the main text (line 45-46) to include the protein mass for yeast and other Eukaryotes.

- Line 49: the sentence starting with “The unique modification profile ...” is difficult to read, please modify.

Action: We have rephrased the sentence in the line 51-52 of the revised main text.

- Fig 1A/Text in lines 94ff: Please add how many reactions were added, and whether they are purely stoichiometric, or whether they account for repeated interaction with client proteins during folding.

Response: All reactions mentioned for protein synthesis are added for client proteins in the model. Each protein may recruit different processes and reactions in the synthesis and folding according to its protein information (such as type and number of the glycosylation sites etc.). The number of purely stoichiometric reactions for producing precursor metabolites for the protein secretory pathway is 92.

Action: We have added one sentence in the legend of Fig. 1a to clarify that those processes are added for production of each protein according to their PTM profile. We have also supplied all added metabolic reactions in Supplementary Table 1 and all reactions and metabolites in pcSecYeast as a Supplementary Data 1.

- Fig 1B: I am astonished that so many proteins are involved in translocation. Does this also include folding chaperones and quality control?

Response: Apologize for the misunderstanding. Fig. 1b depicts the client protein numbers processed in each subsystem. The bar plot for *Transloc* represents that there are over 400 client proteins translocated from cytosol into the ER. In the revised Fig. 1b, we have added the number of machinery proteins in each subsystem.

Action: We have replotted Fig. 1b using the stacked bar plot to specify the machinery proteins and client proteins in each subsystem. We have also added the underlying reason for the large number of proteins involved in the translocation, COPII and ERAD in the legend of Fig. 1b.

- Lines 138ff: Difficult to read, please split the sentence starting with “Thus, at low specific growth rates...”

Action: We have rephrased the whole section in the revised main text (line 150-175).

- Figure 2 and related text: I agree that it is possible to calculate the “secretory cost” of a transporter, but I do not see any evidence that this is truly the driving force that determines which transporter is used. The hypothesis should be proven with experimental data that indeed the Hxt7 transporter is less abundant at low glucose than Hxt1/3 at high glucose, or that overexpression of Hxt7 is more stressful for the cells than overexpression of Hxt1/3 (either by the authors or by literature data).

Response: Thanks for this comment! We rephrased the section to clarify the calculation and the result. To be noted, “Secretory cost” for glucose transporters quantifies the cost for utilizing different glucose transporters to sustain the current growth rate. According to Eq. 1 in the main text, both growth uptake (related with growth rate) and glucose concentration (impacting on the enzyme kinetics) can affect the transporter abundance. Therefore, we cannot assure that at low glucose concentration, the glucose transporter abundance is lower than that at high glucose concentration. It is also the same for the overexpression experiment. Even though it is the case that overexpression of Hxt7 requires high cost, different enzyme kinetics can affect the glucose uptake rates and then the growth rate, therefore, the maximal growth rate cannot be a signal to indicate the stress of glucose transporter expression. Besides that, cells are under different pressure under different glucose concentration. At low glucose concentration, the glucose is the major limitation, while at high glucose concentration, the limited proteome is the major limitation. The cell stress caused by the same amount of glucose transporter abundance under different conditions can therefore be different. Therefore, neither a protein abundance comparison nor overexpression result can support our conclusion.

Out of curiosity, we tried to compare the abundance of Hxt7 at low glucose and Hxt1 at high glucose. We searched absolute proteome/mRNA datasets for *S. cerevisiae*^{28–32}. However, there are few data for the conditions with different glucose concentrations. Besides that, Hxt1 and Hxt7 are membrane proteins, that are often not measured in proteome datasets due to the difficulties in the detection of membrane proteins. We therefore did not manage to get the abundances of these two proteins from the same study (Hxt1 from high glucose and Hxt7 from low glucose). We only observed that Hxt1 increases its abundance from glucose-limited condition to unlimited condition³³ and that HXT1 is more abundant than HXT7 at glucose-unlimited conditions^{34–37}, while on the other hand Hxt7 are present mostly in the glucose-limited conditions³⁰.

Therefore, we have rephrased in the main text that we successfully identified consistency in the secretory cost change with the reported glucose transporter switch, instead of claiming that it is the driving force.

The switch of transporters has been seen in multiple organisms¹²⁻¹⁴. The benefit of utilizing high-affinity transporters during nutrient depletion or limited conditions seems evident, but there remains question why the cell would switch to low affinity transporters. There are several hypotheses for the switch including: 1) the enzyme binding constants, which is based on the rate-affinity trade-offs among different conditions¹⁵. Cells would choose high k_{cat} but low affinity transporters when substrates are abundant; 2) utilization of the low affinity transporter allows cells to sense the substrate depletion accurately and quickly, which would give the cells more time to prepare for the substrate depletion condition¹⁶; and 3) utilization of the low-affinity transporters lowers the efflux of substrates, which contributes higher net import flux of the substrate¹⁷. Each of these hypotheses cannot perfectly explain the benefit of the low affinity transporters. The discussion of these hypotheses can be found in the reference¹⁷, in which they also mentioned that it is hard to perform the experimental evaluation for those three hypotheses.

In this paper, we came up with another new hypothesis, i.e., the secretory cost differences among those transporters, which is not conflicting with other hypothesis. From Eq. 1 in the main text, we can see that the trade-off hypothesis is included in the calculation for secretory cost. We here focus on the model potential in energy cost quantification of utilization different glucose transporters.

Action: We have revised the description in the main text (line 137, line 140-175).

- Fig 2B and C: Currently the secretory cost or protein abundance seems to be inferred from the qGlc. It remains unclear to me why the secretory costs change with glucose concentration, the “unit costs” should be independent of such an external condition. Also, why are costs of Hxt1/3 higher at low glucose?

Response: We are sorry for not clarifying this. There are two main costs defined in the study related to this question: the “unit secretory cost” and “the secretory cost”. The “unit secretory cost” is independent of external conditions (Supplementary Table 3), while “the secretory cost” for glucose transporters combines the “unit secretory cost”, the enzyme kinetics, and the total glucose uptake. “The secretory cost” for each transporter refers to the synthesizing and secretory cost of using the sole corresponding transporter to uptake the total amount required glucose and to sustain a certain specific growth rate. Thus, we can monitor and compare the cost change as we did in Fig. 2c.

“The secretory cost” of Hxt1/3 is higher due to the low affinity of these transporters. Thus, at low glucose concentration, the abundance of Hxt1/3 is higher than Hxt7 if those transporters are used to achieve the same glucose uptake, which leads to a higher cost.

Action: We have rephrased the lines (150-175) in the main text and added a plot in Fig. 2b to indicate that the total uptake rate is used to calculate the secretory cost.

- Looking at the data in Table S3, the results in Figure 2C even get less comprehensible to me: the direct costs and unit costs of Hxt7 and Hxt3 are almost equal (834 vs 836 mol glucose/mol protein direct costs; 3347 vs 3349 mol glucose/mol protein unit costs), also for Hxt1 there is not a huge difference (786 and 3261 mol/mol, respectively). Please explain in more detail how this can lead to the differences observed in Fig 2B, and check carefully if a circular argument might have been generated, through the calculation of protein abundance via its catalytic properties in Eq 1 rather than using protein abundance from proteomic resources.

Response: Even though the difference in ‘unit secretory cost’ is very small, considering the high glucose uptake after switching to aerobic fermentation at high growth rates, the cost difference actually matters.

Fig. 2c shows “the secretory cost” rather than ‘the unit secretory cost’. The secretory cost refers to the cost for using different sole glucose transporter to sustain corresponding growth rate under the external glucose concentration, which is the combination result of enzyme kinetics, “the unit secretory cost”, external glucose concentration and total glucose uptake rate.

We have carefully checked the published proteome data as discussed in response to previous comments for Fig. 2, there is not such available data with absolute Hxt1, Hxt3 and Hxt7 protein abundance measurement under different growth rate conditions from the same study.

Action: We have rephrased the lines (150-175) in the main text and added a subplot in Fig 2b to indicate the total glucose uptake rate is used to calculate the secretory cost.

- Please add the systematic names of all genes/proteins that are discussed in the text (or add the short names to the supplementary tables S3 and S4).

Response: Thanks for this comment! We have updated the systematic names in the Supplementary Table 3 and 4.

Action: We have updated the Supplementary Tables.

- The text in lines 125-161 is rather hard to read, sometimes sentences are connected without a connecting word, sometimes words are missing or in the wrong grammatical form.

Action: We have carefully rewritten the corresponding lines (150-175) in the revised main text.

- The results described and discussed in Lines 164-193 are highly interesting, even though not unexpected. It might be interesting to look a bit more in detail which are the most costly secretory proteins (e.g. if there are some characteristic features or PTMs associated with being costly).

Response: Thanks for this comment. The costliest secretory proteins (top 30) have an average 1500 amino acid length, which is three time of the average length (467 amino acids) of *S. cerevisiae* proteins. The long protein increases not only the “direct cost” but also the energy required for the translocation and folding in the secretory pathway. Besides that, 17 out of these 30 proteins contain multiple *N*-glycosylation sites (average 11 *N*-glycosylation sites of those 30 proteins), which also contribute to the high cost.

Action: We have added the PTM and protein length in the Supplementary Table 4 for all secretory proteins to facilitate the understanding.

- Supplementary Figure 2b: It might be more convincing to show the correlation of one of the two higher producers rather than of AAC1.

Response: Sorry we made a small mistake in this figure. The figure is for the higher producer MH34 rather than AAC1.

Action: We have revised the legend.

- Why didn't the authors do the correlation to the proteome data they published previously (Qi et al. 2020. mBio 11(6):e02743-20)? Is this because they wanted to show that also in yeast transcriptional repression of secretory proteins is occurring as described for mammalian cells and filamentous fungi (Pakula et al. 2003. J Biol Chem. 278(45):45011-20)?

Response: Thanks for this concern. We tried to do the correlation using the proteome data, but we were challenged by the fact that a large fraction of secretory proteins is localized in the cell membrane, which

are known to be hard to detect. In our previous study³⁸, we detected more than 2800 proteins in the proteome, but only 167 out of 497 secretory proteins were detected, which are quite low coverage compared with the number of detected mRNAs in the transcriptome data (463 out of 497). On the other hand, we think that transcription is a more direct response towards the pressure of recombinant protein production as the reviewer suggested here. Based on above reasons, we decided to do the correlation to the transcriptome data rather than the proteome data.

- Line 199: It would be good to include at least one yeast reference here (e.g. instead of ref 24 dealing with ER stress in the heart). The recent review by Ninagawa et al. 2021 *Biochim Biophys Acta Gen Subj.* 1865(3):129812 might be a good option covering both yeast and mammalian systems.

Action: We have added the reference in the revised main text (line 214).

- Regarding protein degradation, the authors might find this interesting: A Systematic Protein Turnover Map for Decoding Protein Degradation. Christiano et al. 2020. *Cell Rep.* 33(6):108378.

Response: Thanks for this reference! We have updated the k_{deg} ratio from this reference into the model. We also identified the mean protein k_{deg} ratio ($\frac{k_{deg}}{k_{deg} + \mu}$) is around 30% in those two studies (In Christiano et al³⁵, mean k_{deg} rate is 0.24 h⁻¹, growth rate is assumed to be 0.4h⁻¹ since it was measured in the exponential growth phase, thus, mean k_{deg} ratio is 35%; in Lahtvee et al²⁸, mean k_{deg} rate is 0.042h⁻¹, growth rate is 0.1 h⁻¹, mean k_{deg} ratio is 30%). Both these two studies are in line with previous report that around 30% nascent peptide are degraded in eukaryotic cells³⁹⁻⁴¹.

Action: We have updated the k_{deg} ratio based on the study by Christiano et al³⁵ and rerun all analyses. All results and figures have been updated. All conclusions remain the same.

- Line 212: I think it should be "...in the ER for different times"

Action: We have fixed this in line 228 in the revised main text.

- Line 217: ...expression of wild type YFP in the cytosol... ... was observed when expressing ...

Action: We have rewritten the section according to the Reviewer 2's comment.

- Line 233: Please rephrase, there seems to be something (at least a word) missing, which makes the meaning of the sentence unclear.

Action: We have rephrased the sentence (line 238-241) in the main text.

- Line 237ff/Supplementary Figure S4: Which reaction catalyzes this flux? To my knowledge GSSG export from the ER is not characterized. However, the predicted flux would fit to the experimental data from Merksamer et al. 2008 *Cell.* 135(5):933-47.

Response: The transport reaction in the model was added into the model for gap-filling purpose, which does not contain any gene annotation so far, since the gene has not been characterized as the reviewer mentioned. The GSSG in the ER can only be reduced in the cytosol, therefore, if there is a high GSSG level in the ER, then it would be reflected by the flux of the transport reaction.

Action: We have added the reference in the revised manuscript (line 249).

- Supplementary Figure S3: Actually, I assume that the increased abundance of Kar2 and Pdi1 in the "misfold-retention" simulations might represent UPR induction. Was activation of this stress response pathway included in the model?

Response: We did not add any UPR induction in our model, the increased abundance of Kar2 and Pdi1 reflects the retention of misfolded protein, since those misfolded protein binds to Kar2 and Pdi1. We added this reaction in the model but not the further stress response, but it can be extended in the future if the UPR and misfolding are of interest. In fact, the model captures that the Kar2 and Pdi1 become the bottleneck if misfolding is increased and we know that the activation of UPR entails the over-expression of Kar2 and Pdi1 and temporarily downregulation of translation to clean up the ER. So, our model does to some extent cover the consequences of the UPR, but of course not the mechanistic process underlying this pathway.

- Figure 3a: There should be reversible arrows for the retention part or a connection to ERAD. I understand that might not be easy or useful to include this in the model, but in vivo ER retention is usually not the final fate in yeast, the majority of misfolded proteins will be eventually degraded, but different misfolded proteins might be retained for different times or accumulate to different abundances, leading to different stress levels as simulated.

Response: Thanks for this comment! Yes, we totally agree with the reviewer that the majority of misfolded proteins will eventually be degraded. Therefore, we added the arrow from the misfolding to the ERAD and denoted in the legend that this reaction is not considered in the model.

Action: We have revised Fig. 3a and its legend.

- Line 244 ff: See also my comment below (model construction), Sec61 seems not to be part of the retro-translocon.

Response: Thanks for this comment. Previously, we included the Sec61 translocon into the constraint for the retro-translocon since Sec61 was found involved in the retro-translocation of misfolded ERAD-L substrates to cytosol^{42,43}. However, other later studies clearly show that there is no interaction of Sec61 with components of other retro-translocation complexes as the reviewer mentioned^{44,45}. In this version, we have removed the Sec61 from the retro-translocation constraint and rerun the analysis.

Action: We have removed the Sec61 from the retro-translocation constraint and rerun the analysis. Fig. 3 has been updated and the conclusion remained the same.

- Line 253: ...in the ER.

Response: We have fixed the text (line 262 and the whole main text).

- Line 269: None of the references 29-31 seems to deal with *Pichia pastoris*, please add (e.g. review by Looser et al. 2015. *Biotechnol Adv.* 33(6 Pt 2):1177-93) or rephrase.

Response: Thanks for this comment! We have added the reference for *Pichia pastoris* (line 278 in the revised main text).

- Figure 6: please add to the legend which promoter/vector was used for overexpression and how many clones per construct were analysed.

Response: The gene fragments mentioned in Fig. 6 were amplified from the yeast genome and assembled into the pSP-GM1 expression vector under the control of *TEF1* promoter, respectively. Data are shown as average values \pm standard errors of three independent biological triplicates. α -amylase was stably expressed on the multi-copy plasmid CPOTud under the control of *TPI1* promoter in a *tpi* deletion background strain.

Action: We have added that information in the legend of Fig. 6.

- Line 352: Overexpression of Ero1 has recently also been predicted to be beneficial for protein production by a very limited model of oxidative folding by Beal et al. 2019. *Antioxid Redox Signal.* 31(4):261-274.

Action: We have added several lines to describe this (line 360-361) and included the reference.

- Line 371: ...is much higher...

Action: We have fixed this in the revised main text (line 384).

- Lines 389-390: I don't think "evolutionary" is the right word here, please rephrase the last part of the sentence.

Action: We have changed it to "evolved through the long history" in text (line 401)

- Line 399: I don't think "recently" fits when something has been postulated more than 10 years ago.

Action: We have removed the "recently".

- Line 428: Please replace "identify" with "predict"

Action: We have fixed this in the revised main text (line 443).

- Line 428: This work definitely is the first comprehensive genome scale model integrating the secretory processes, and thus also the first one to predict engineering targets in these processes. However, engineering targets for recombinant protein production have been predicted by GSMMs before, albeit not for secretory functions. I recommend to rephrase this final paragraph accordingly, and give credits to these earlier works (either here or in the introduction), that improved protein production by model predictions in various host organisms.

Response: Thanks for pointing this out! We have added sentences in the introduction to describe the earlier work and revised the final paragraph.

Action: We have added sentences in the introduction (line 77-80) and revised the final paragraph (440-443).

Model construction:

- Signal peptidase should also be involved in post-translational translocation (reading further I see it is introduced as a separate topic later, but it might be good to restructure this in the text)

Response: We are sorry for causing the misunderstanding. We have revised the method part.

Action: We have added sentences and restructured this part (line 200-201 in the Supplementary Methods).

- Why are the Golgi enzymes not considered in N-glycosylation? (reading further I see it is introduced as a separate topic later, but it might be good to restructure this in the text)

Action: We have added sentences in this part (line 316-318 in the Supplementary Methods).

- Literature suggests that the Sec61 translocon is not involved in ERAD. Instead Hrd1/Der1 were found to form the channel for retrotranslocation (e.g. Wu et al. 2020 *Science*. 368(6489):eaaz2449.)

Response: Thanks for this comment. We included the Sec61 translocon into the constraint for the retro-translocon since Sec61 was found involved in the retro-translocation of misfolded ERAD-L substrates to cytosol^{42,43}. However, other later studies clearly show that there is no interaction of Sec61 with components of other retro-translocation complexes as the reviewer mentioned^{44,45}. In this version, we have removed the Sec61 from the retro-translocation constraint and rerun the analysis.

Action: We have removed the Sec61 from the retro-translocation constraint and rerun the analysis. Fig. 3 has been updated and the conclusion has remained.

- Doa10 seems to be the ubiquitin ligase also for ERAD-M, not only ERAD-C (Schmidt et al. 2020. Elife. 9:e56945.).

Response: Thanks for this comment! We have added an alternative pathway which utilizes Doa10 for ERAD-M in the model.

Action: We have rebuilt the model and rerun all analyses combined with changes mentioned in response to last comment.

- Whole manuscript and supplements: “detailly” is not recognized by any of the authoritative English dictionaries. The closest recognized term is in detail.

Action: We have revised the word to “in detail” and “comprehensively”

- The supplement detailing the model construction should be proof-read, there are several typos including one in the model itself (tanslocate instead of translocate)

Action: We have proof-read the Supplementary Methods and revised the typo in the supplementary methods, the code and the model.

- Whole manuscript and supplements: The quality of English is varying between different passages. This needs to be amended, either by one of the authors or a native speaker.

Response: Thanks! We have proof-read and revised the whole manuscript and the supplements.

References:

1. Elsemman, I. E. *et al.* Whole-cell modeling in yeast predicts compartment-specific proteome constraints that drive metabolic strategies. *Nat. Commun.* **13**, 801 (2022).
2. Oftadeh, O. *et al.* A genome-scale metabolic model of *Saccharomyces cerevisiae* that integrates expression constraints and reaction thermodynamics. *Nat. Commun.* **12**, 4790 (2021).
3. Ye, C. *et al.* Comprehensive understanding of *Saccharomyces cerevisiae* phenotypes with whole-cell model WM_S288C. *Biotechnol. Bioeng.* **117**, 1562–1574 (2020).
4. Irani, Z. A., Kerkhoven, E. J., Shojaosadati, S. A. & Nielsen, J. Genome-scale metabolic model of *Pichia pastoris* with native and humanized glycosylation of recombinant proteins. *Biotechnol. Bioeng.* **113**, 961–969 (2016).
5. Gutierrez, J. M. *et al.* Genome-scale reconstructions of the mammalian secretory pathway predict metabolic costs and limitations of protein secretion. *Nat. Commun.* **11**, 68 (2020).
6. Lloyd, C. J. *et al.* COBRAme : A computational framework for genome-scale models of metabolism and gene expression. 1–14 (2018).
7. Woolford, J. L. J. & Baserga, S. J. Ribosome biogenesis in the yeast *Saccharomyces cerevisiae*. *Genetics* **195**, 643–681 (2013).
8. de la Cruz, J., Karbstein, K. & Woolford, J. L. Functions of ribosomal proteins in assembly of eukaryotic ribosomes in vivo. *Annu. Rev. Biochem.* **84**, 93–129 (2015).

9. Chen, Y. *et al.* Proteome constraints reveal targets for improving microbial fitness in nutrient-rich environments. *Mol. Syst. Biol.* **17**, e10093 (2021).
10. Sánchez, B. J. *et al.* Improving the phenotype predictions of a yeast genome-scale metabolic model by incorporating enzymatic constraints. *Mol. Syst. Biol.* **13**, 935 (2017).
11. Chen, Y., Li, F., Mao, J., Chen, Y. & Nielsen, J. Yeast optimizes metal utilization based on metabolic network and enzyme kinetics. *Proc. Natl. Acad. Sci.* **118**, (2021).
12. Thorens, B. & Mueckler, M. Glucose transporters in the 21st Century. *Am. J. Physiol. Endocrinol. Metab.* **298**, E141-5 (2010).
13. Castro, R. *et al.* Characterization of the individual glucose uptake systems of *Lactococcus lactis*: mannose-PTS, cellobiose-PTS and the novel GlcU permease. *Mol. Microbiol.* **71**, 795–806 (2009).
14. Diderich, J. A. *et al.* Glucose uptake kinetics and transcription of HXT genes in chemostat cultures of *Saccharomyces cerevisiae*. *J. Biol. Chem.* **274**, 15350–15359 (1999).
15. Gudelj, I., Beardmore, R. E., Arkin, S. S. & MacLean, R. C. Constraints on microbial metabolism drive evolutionary diversification in homogeneous environments. *J. Evol. Biol.* **20**, 1882–1889 (2007).
16. Levy, S., Kafri, M., Carmi, M. & Barkai, N. The competitive advantage of a dual-transporter system. *Science* **334**, 1408–1412 (2011).
17. Bosdriesz, E. *et al.* Low affinity uniporter carrier proteins can increase net substrate uptake rate by reducing efflux. *Sci. Rep.* **8**, 5576 (2018).
18. Huang, M. *et al.* Microfluidic screening and whole-genome sequencing identifies mutations associated with improved protein secretion by yeast. *Proc. Natl. Acad. Sci. U. S. A.* **112**, E4689-96 (2015).
19. Bao, J., Huang, M., Petranovic, D. & Nielsen, J. Moderate Expression of SEC16 Increases Protein Secretion by *Saccharomyces cerevisiae*. *Appl. Environ. Microbiol.* **83**, (2017).
20. Huang, M., Bao, J., Hallström, B. M., Petranovic, D. & Nielsen, J. Efficient protein production by yeast requires global tuning of metabolism. *Nat. Commun.* **8**, 1131 (2017).
21. Wang, M., Herrmann, C. J., Simonovic, M., Szklarczyk, D. & von Mering, C. Version 4.0 of PaxDb: Protein abundance data, integrated across model organisms, tissues, and cell-lines. *Proteomics* **15**, 3163–3168 (2015).
22. Stolz, A. & Wolf, D. H. Use of CPY and its derivatives to study protein quality control in various cell compartments. *Methods Mol. Biol.* **832**, 489–504 (2012).
23. Haynes, C. M., Titus, E. A. & Cooper, A. A. Degradation of misfolded proteins prevents ER-derived oxidative stress and cell death. *Mol. Cell* **15**, 767–776 (2004).
24. Glembotski, C. C. Endoplasmic reticulum stress in the heart. *Circ. Res.* **101**, 975–984 (2007).
25. Gregersen, N., Bross, P., Vang, S. & Christensen, J. H. Protein misfolding and human disease. *Annu. Rev. Genomics Hum. Genet.* **7**, 103–124 (2006).
26. Ninagawa, S., George, G. & Mori, K. Mechanisms of productive folding and endoplasmic reticulum-associated degradation of glycoproteins and non-glycoproteins. *Biochim. Biophys. Acta. Gen. Subj.* **1865**, 129812 (2021).
27. Chen, Y. & Nielsen, J. Mathematical modelling of proteome constraints within metabolism. *Curr. Opin. Syst. Biol.* (2021).
28. Lahtvee, P.-J. *et al.* Absolute quantification of protein and mRNA abundances

- demonstrate variability in gene-specific translation efficiency in yeast. *Cell Syst.* **4**, 495–504 (2017).
29. Yu, R., Vorontsov, E., Sihlbom, C. & Nielsen, J. Quantifying absolute gene expression profiles reveals distinct regulation of central carbon metabolism genes in yeast. *Elife* **10**, (2021).
 30. Yu, R. *et al.* Nitrogen limitation reveals large reserves in metabolic and translational capacities of yeast. *Nat. Commun.* **11**, 1881 (2020).
 31. Di Bartolomeo, F. *et al.* Absolute yeast mitochondrial proteome quantification reveals trade-off between biosynthesis and energy generation during diauxic shift. *Proc. Natl. Acad. Sci. U. S. A.* **117**, 7524–7535 (2020).
 32. Ho, B., Baryshnikova, A. & Brown, G. W. Unification of Protein Abundance Datasets Yields a Quantitative *Saccharomyces cerevisiae* Proteome. *Cell Syst.* 1–14 (2018). doi:10.1016/j.cels.2017.12.004
 33. Diderich, J. A. *et al.* Glucose uptake kinetics and transcription of HXT genes in chemostat cultures of *Saccharomyces cerevisiae*. *J. Biol. Chem.* **274**, 15350–15359 (1999).
 34. Björkeröth, J. *et al.* Proteome reallocation from amino acid biosynthesis to ribosomes enables yeast to grow faster in rich media. *Proc. Natl. Acad. Sci. U. S. A.* **117**, 21804–21812 (2020).
 35. Christiano, R. *et al.* A Systematic Protein Turnover Map for Decoding Protein Degradation. *Cell Rep.* **33**, 108378 (2020).
 36. Ghaemmaghami, S. *et al.* Global analysis of protein expression in yeast. *Nature* **425**, 737–741 (2003).
 37. Malina, C., Yu, R., Björkeröth, J., Kerkhoven, E. J. & Nielsen, J. Adaptations in metabolism and protein translation give rise to the Crabtree effect in yeast. *Proc. Natl. Acad. Sci.* **118**, e2112836118 (2021).
 38. Qi, Q. *et al.* Different Routes of Protein Folding Contribute to Improved Protein Production in *Saccharomyces cerevisiae*. *MBio* **11**, (2020).
 39. Eisenlohr, L. C., Huang, L. & Golovina, T. N. Rethinking peptide supply to MHC class I molecules. *Nat. Rev. Immunol.* **7**, 403–410 (2007).
 40. Qian, S.-B., Princiotta, M. F., Bennink, J. R. & Yewdell, J. W. Characterization of rapidly degraded polypeptides in mammalian cells reveals a novel layer of nascent protein quality control. *J. Biol. Chem.* **281**, 392–400 (2006).
 41. Ha, S.-W., Ju, D., Hao, W. & Xie, Y. Rapidly Translated Polypeptides Are Preferred Substrates for Cotranslational Protein Degradation. *J. Biol. Chem.* **291**, 9827–9834 (2016).
 42. Römisch, K. A Case for Sec61 Channel Involvement in ERAD. *Trends Biochem. Sci.* **42**, 171–179 (2017).
 43. Plemper, R. K. *et al.* Genetic interactions of Hrd3p and Der3p/Hrd1p with Sec61p suggest a retro-translocation complex mediating protein transport for ER degradation. *J. Cell Sci.* **112 (Pt 2)**, 4123–4134 (1999).
 44. Mehnert, M., Sommer, T. & Jarosch, E. Der1 promotes movement of misfolded proteins through the endoplasmic reticulum membrane. *Nat. Cell Biol.* **16**, 77–86 (2014).
 45. Carvalho, P., Stanley, A. M. & Rapoport, T. A. Retrotranslocation of a misfolded luminal ER protein by the ubiquitin-ligase Hrd1p. *Cell* **143**, 579–591 (2010).

Reviewers' Comments:

Reviewer #1:

Remarks to the Author:

The authors have satisfactorily addressed all of my concerns.

Reviewer #2:

Remarks to the Author:

The authors have addressed most of the concerns. But some of these changes are not reflected in the manuscript.

Also response to the comment by Rev2 "It is also not clear how 17 targets were chosen for this. A high-level overview without specific numbers is provided in the methods section. No quantitative comparison was made for these results as well, such as through correlation, precision, recall, hypergeometric tests or AUC values." was not addressed adequately.

Only a single sentence was added to address this in line 591-593. "After the priority rank, we clustered those proteins based on their functions and selected these 18 proteins with different functions and ranked with high priority score for validation";

this is inadequate; potentially a histogram showing the distribution of predictions and highlighting those that were selected might be useful. the clustergram that was used for selecting these could also be shown.

The Rev2 comment above did not expect nor request them to experimentally test all 1639 genes. The comment requested them to perform a quantitative comparison of the data already presented, which has not yet been performed. The data from figure 6b 6c could be used to create a scatter plot of predictions vs experiments and correlations can be determined to assess quantitative fit. Currently only the experimental results are shown and are described qualitatively. There's no statistics on how many genes were predicted to increase production and how many were chosen for experiments.

It would be useful to have 'Table R1. Comparison of pcSecYeast with other models' in some form in the main text since it was raised by two reviewers and I think it greatly clears up the novelty and contribution of this current study.

Response to Rev 1 Q2a & 2b should also be included in the manuscript/discussion or methods

Response to Rev 2 comment 'The conclusion that misfolding and retention imposes greater fitness costs than correct folding and degradation respectively is obvious again.' should also be discussed in the manuscript.

Reviewer #3:

Remarks to the Author:

The authors have answered to all my comments and concerns, and made the appropriate changes to the manuscript and the supplements.

REVIEWER COMMENTS

Reviewer #1 (Remarks to the Author):

The authors have satisfactorily addressed all of my concerns.

Reviewer #2 (Remarks to the Author):

The authors have addressed most of the concerns. But some of these changes are not reflected in the manuscript.

Response: Thanks for pointing this out! We have revised the manuscript accordingly.

Also response to the comment by Rev2 "It is also not clear how 17 targets were chosen for this. A high-level overview without specific numbers is provided in the methods section. No quantitative comparison was made for these results as well, such as through correlation, precision, recall, hypergeometric tests or AUC values." was not addressed adequately.

Only a single sentence was added to address this in line 591-593. "After the priority rank, we clustered those proteins based on their functions and selected these 18 proteins with different functions and ranked with high priority score for validation";

this is inadequate; potentially a histogram showing the distribution of predictions and highlighting those that were selected might be useful. the clustergram that was used for selecting these could also be shown.

Response: We are sorry for this. In total, we have predicted 116 overexpression targets for α -amylase overproduction from FSEOF. In order to select predicted targets for further validation, we adopted several criteria to rank the priority of those targets. We checked whether it is in a multi-subunits complex (Fig. R1a), whether it have paralogs (Fig. R1b), whether the protein abundance is sufficient *in vivo* (calculated from the simulated candidate protein abundance and the average measured protein abundance of PaxDb¹, Fig. R1c) whether the ratio of protein abundance has not changed significantly from low production of α -amylase to high production (Fig. R1d). The above-mentioned scenarios may hinder the overexpression performance, which are therefore ranked with low priority scores. After the rank, we grouped the proteins based on their functions and selected the 18 targets with different functions for validation (Fig. R2). Most of them were ranked with high priority scores except CRS1(Fig. R1e and Fig. R2). Since there is no previous report about how tRNA synthetase could affect secretory protein production, and we are curious about the effect, CRS1 was also added into the validation.

Action: We have added several lines (lines 331-334 in the revised manuscript) and Supplementary Figure 8 to address this issue.

Fig. R1 Parameters of predicted overexpression targets in amylase overproduction. (a) *Non subunit protein* represents that the target is not a subunit in an enzyme complex. (b) *Non paralog exist* represents that there is no paralog for the target protein in the genome. (c) *Log10 fold change (FC) of simulated target protein abundance compared with PaxDb*, which indicates the changes in target protein abundance from the average measured proteome abundance to high α -amylase production condition. Simulated protein abundance is the outcome of the model simulations on high α -amylase production state. (d) *Fold change (FC) of simulated target protein abundance* represents the ratio of predicted target protein abundance at high α -amylase production condition over that in low amylase production condition. Abundances for these two states are both from the model simulation. (e) Priority score represents the combination score from evaluation of the previous four parameters. Target proteins with high priority score should be prioritized for experiments.

Fig. R2. The pathway analysis for the predicted total 116 overexpression targets and the selected 18 targets. Cys: Cysteine synthesis, tRNA: tRNA ligase, Oligo: oligosaccharide biosynthesis, ER_Golgi trans: ER to Golgi transport, Cytosol_ER transloc: Cytosol to ER translocation, NG: N-glycosylation, DSB: disulfide bond related pathway, ERAD: Endoplasmic reticulum (ER)-associated protein degradation. The color of the outer circle represents the priority score for the candidate. Selected targets are denoted in the figure.

The Rev2 comment above did not expect nor request them to experimentally test all 1639 genes. The comment requested them to perform a quantitative comparison of the data already presented, which has not yet been performed. The data from figure 6b 6c could be used to create a scatter plot of predictions vs experiments and correlations can be determined to assess quantitative fit. Currently only the experimental results are shown and are described qualitatively. There's no statistics on how many genes were predicted to increase production and how many were chosen for experiments.

Response: In total 116 genes were predicted as targets, and we have chosen 18 for further validation.

For the quantitative comparison, we assume that the reviewer wants us to plot a figure with $\frac{\text{experimental FC}(\alpha\text{-amylase})}{\text{experimental FC}(\text{abundance of target protein})}$ versus $\frac{\text{predicted FC}(\alpha\text{-amylase})}{\text{predicted FC}(\text{abundance of target protein})}$. FC stands for

fold change. We agree with the reviewer that having such a figure would be great, but the current data is not sufficient for such a plot.

As for the experimental part, we have measured experimental FC(α -amylase), but we do not know the experimental fold change of abundances of tested 14 target proteins. Since we used a multi-copy plasmid (2micro) to overexpress the target genes, the expression levels vary between genes, which are not available.

As for the predicted part, we cannot calculate $\frac{\text{predicted FC}(\alpha\text{-amylase})}{\text{predicted FC}(\text{abundance of target protein})}$ from the FSEOF results. Since the FSEOF only predicts *fold changes of abundances of all native proteins (including target proteins)* resulted from the enforcement increase of α -amylase production rate (Fig. R3). FC(α – amylase) is the enforcement input for the FSEOF, rather than the outcome. The goal of FSEOF is to determine the limiting proteins in terms of α -amylase overproduction rather than to quantify the effect of overexpression of a single gene. Thus, FSEOF cannot be used to calculate $\frac{\text{predicted FC}(\alpha\text{-amylase})}{\text{predicted FC}(\text{abundance of target protein})}$.

Moreover, we checked metabolic engineering literatures to search for a method to perform the quantitative comparison requested by the reviewer, and found that the studies using FSEOF also did not quantitatively compare predictions with experimental validations²⁻⁵. We also checked the model-guided metabolic engineering studies using other algorithms, and found that most of them also did not perform such quantitative correlation analysis⁶⁻¹² with only a few exceptions, which however did not show perfect correlation^{13,14}. Therefore, it is not common to perform the quantitative comparison between model predictions with experimental validations for metabolic engineering targets as predicting effective targets that cannot be predicted by other existing models is already a significant improvement.

The major contribution of pcSecYeast is to systematically predict the potential targets in the complex secretory pathway for engineering in terms of recombinant protein overproduction. In this study, several overexpression targets for α -amylase overproduction were identified and validated for the first time. This shows the power of pcSecYeast to identify the potential targets.

Action: We added the number of genes that were predicted to increase production and number of genes that were chosen for experiments (lines 331-334 in the revised main text). We also rewrote the methods part for the FSEOF to explain the prediction part (lines 590-599 in the revised manuscript). We also added some explanation in the Fig. 5a legend to facilitate the understanding of the method.

Fig. R3 Adapted FSEOF method for target identification. Firstly, we reduced the specific growth rate in the simulation. The carbon flow towards the growth can be diverted to the recombinant protein production by maximizing the recombinant protein production. Then from the simulated native protein abundances, we can select those proteins with simulated abundance increase resulted from the enforcement of recombinant protein production as initial targets. Priority rank was then performed to further select the targets.

It would be useful to have 'Table R1. Comparison of pcSecYeast with other models' in some form in the main text since it was raised by two reviewers and I think it greatly clears up the novelty and contribution of this current study.

Action: Thanks for this suggestion! We have formulated a Table 1 and one sentence (line 102-104 in the revised main text). We hope that this would clarify the difference of pcSecYeast with previous models.

Response to Rev 1 Q2a & 2b should also be included in the manuscript/discussion or methods

Action: We have added this part in the Methods section (lines 473-482 in the revised manuscript).

Response to Rev 2 comment 'The conclusion that misfolding and retention imposes greater fitness costs than correct folding and degradation respectively is obvious again.' should also be discussed in the manuscript.

Action: Thanks for this suggestion! We have added this part in the discussion section (lines 411-418 in the revised manuscript).

Reviewer #3 (Remarks to the Author):

The authors have answered to all my comments and concerns, and made the appropriate changes to the manuscript and the supplements.

References

1. Wang, M., Herrmann, C. J., Simonovic, M., Szklarczyk, D. & von Mering, C. Version 4.0 of PaxDb: Protein abundance data, integrated across model organisms, tissues, and cell-lines. *Proteomics* **15**, 3163–3168 (2015).
2. Wang, X., Yu, L. & Chen, S. UP Finder: A COBRA toolbox extension for identifying gene overexpression strategies for targeted overproduction. *Metab. Eng. Commun.* **5**, 54–59 (2017).
3. Choi, H. S., Lee, S. Y., Kim, T. Y. & Woo, H. M. In silico identification of gene amplification targets for improvement of lycopene production. *Appl. Environ. Microbiol.* **76**, 3097–3105 (2010).
4. Badri, A., Raman, K. & Jayaraman, G. Uncovering Novel Pathways for Enhancing Hyaluronan Synthesis in Recombinant *Lactococcus lactis*: Genome-Scale Metabolic Modeling and Experimental Validation. *Processes* **7**, (2019).
5. Kim, M. *et al.* Reconstruction of a high-quality metabolic model enables the identification of gene overexpression targets for enhanced antibiotic production in *Streptomyces coelicolor* A3(2). *Biotechnol. J.* **9**, 1185–1194 (2014).
6. Ng, C., Jung, M., Lee, J. & Oh, M.-K. Production of 2,3-butanediol in *Saccharomyces cerevisiae* by in silico aided metabolic engineering. *Microb. Cell Fact.* **11**, 68 (2012).
7. Kim, K. *et al.* Engineering *Bacteroides thetaiotaomicron* to produce non-native butyrate based on a genome-scale metabolic model-guided design. *Metab. Eng.* **68**, 174–186 (2021).
8. Alper, H., Jin, Y.-S., Moxley, J. F. & Stephanopoulos, G. Identifying gene targets for the metabolic engineering of lycopene biosynthesis in *Escherichia coli*. *Metab. Eng.* **7**, 155–164 (2005).
9. Alper, H., Miyaoku, K. & Stephanopoulos, G. Construction of lycopene-overproducing *E. coli* strains by combining systematic and combinatorial gene knockout targets. *Nat. Biotechnol.* **23**, 612–616 (2005).

10. Hwan, P. J., Ho, L. K., Yong, K. T. & Yup, L. S. Metabolic engineering of *Escherichia coli* for the production of l-valine based on transcriptome analysis and in silico gene knockout simulation. *Proc. Natl. Acad. Sci.* **104**, 7797–7802 (2007).
11. Xu, P., Ranganathan, S., Fowler, Z. L., Maranas, C. D. & Koffas, M. A. G. Genome-scale metabolic network modeling results in minimal interventions that cooperatively force carbon flux towards malonyl-CoA. *Metab. Eng.* **13**, 578–587 (2011).
12. Ranganathan, S. *et al.* An integrated computational and experimental study for overproducing fatty acids in *Escherichia coli*. *Metab. Eng.* **14**, 687–704 (2012).
13. Asadollahi, M. A. *et al.* Enhancing sesquiterpene production in *Saccharomyces cerevisiae* through in silico driven metabolic engineering. *Metab. Eng.* **11**, 328–334 (2009).
14. Boghigian, B. A., Armando, J., Salas, D. & Pfeifer, B. A. Computational identification of gene over-expression targets for metabolic engineering of taxadiene production. *Appl. Microbiol. Biotechnol.* **93**, 2063–2073 (2012).

Reviewers' Comments:

Reviewer #2:

Remarks to the Author:

The authors have done a good job addressing all my comments. There are no additional concerns.